# Eikonal amplitudes from curved backgrounds

**Tim Adamo,**[1] **Andrea Cristofoli,**[1] **Piotr Tourkine**[2]

[1]*School of Mathematics and Maxwell Institute for Mathematical Sciences*
*University of Edinburgh, EH9 3FD, United Kingdom*
[2]*LAPTh, CNRS et Université Savoie Mont-Blanc, 9 Chemin de Bellevue, F-74941 Annecy, France*

ABSTRACT: Eikonal exponentiation in QFT describes the emergence of classical physics at long distances in terms of a non-trivial resummation of infinitely many diagrams. Long ago, 't Hooft proposed a beautiful correspondence between ultra-relativistic scalar eikonal scattering and one-to-one scattering in a background shockwave space-time, bypassing the need to resum. In this spirit, we propose a covariant method for computing one-to-one amplitudes in curved background space-times which gives rise what we conjecture to be a general expression for the eikonal amplitude. We show how the one-to-one scattering amplitude for scalars on any stationary space-time reduces to a boundary term that captures the long-distance behavior of the background and has the structure of an exponentiated eikonal amplitude. In the case of scalar scattering on Schwarzschild, we recover the known results for gravitational scattering of massive scalars in the eikonal regime. For Kerr, we find a remarkable exponentiation of the tree-level amplitude for gravitational scattering between a massive scalar and a massive particle of infinite spin. This amplitude exhibits a Kawai-Lewellen-Tye-like factorization, which we use to evaluate the eikonal amplitude in momentum space, and study its analytic properties.

## 1 Introduction

The eikonal approximation is a well-known and powerful tool in quantum field theory [1–8]. For some theories, in the high-energy regime of $2 \to 2$ scattering with small momentum transfer (i.e. $-t \ll s$ in terms of the Mandelstam variables), the leading dominant diagrams are ladders and crossed ladders whose rungs are exchanges of the highest available spin $J$ in the theory ($J = 1, 2$ for photon or graviton interactions). For gravitationally-coupled theories, this means that ladder diagrams with graviton exchanges dominate [9]. This infinite series of ladder diagrams can be resummed into an *eikonal amplitude* of remarkable simplicity. This eikonal amplitude is controlled by the *eikonal phase*, which at leading order in the eikonal expansion is given by the inverse transverse Fourier transform of the single exchange diagram or Born amplitude.

The simplicity of eikonal amplitudes and the circumstances under which eikonal exponentiation holds have been studied in recent years in several contexts, from $\mathcal{N} = 8$

supergravity [10–12] to applications in classical gravitational wave physics [13–31]. Interestingly, while eikonal exponentiation of quantum scattering amplitudes is known to hold in some cases and to fail for others (e.g., pure cubic scalar theory [32–34]), there are still settings where its status is not entirely explored. An example of this is the actual evaluation of the eikonal amplitude for $2 \rightarrow 2$ scattering of massive particles with arbitrarily large quantum spin [35–39], which has been recently investigated in [40].

A beautiful explanation for eikonal exponentiation and the classicality of eikonal scattering was provided long ago by 't Hooft [9] in the ultra-relativistic limit of graviton-mediated scalar scattering[1]. He observed that in the ultra-relativistic limit, each incoming scalar sees the other as a strong, ultraboosted source following a light-like trajectory. Such a source is described exactly in general relativity by the shockwave solution [42, 43]. The $2 \rightarrow 2$ scattering process is then recast as semi-classical $1 \rightarrow 1$ scattering of a massless scalar in the fixed, non-perturbative background of the shockwave. This correspondence between ultra-relativistic eikonal and shockwave scattering has played a major role in the study of transplanckian scattering in quantum gravity (e.g., [44–49]), and also holds in theories like QED (cf., [50, 51]). However, there remains no clear correspondence between eikonal amplitudes and more general curved backgrounds[2].

The main idea of this paper is to provide a framework for computing generic eikonal amplitudes in terms of scattering on curved background space-times, generalizing the original proposal of 't Hooft [9]. In particular, one can view the shockwave calculation as a special case of a more general phenomenon: a correspondence between gravitationally-mediated eikonal $2 \rightarrow 2$ scattering on the one hand, and $1 \rightarrow 1$ scattering on a classical, curved space-time whose source corresponds to one of the scattering states in the eikonal picture. This opens the door to computing eikonal amplitudes directly from scattering in curved space-times, rather than by resumming the ladder diagrams of the high-energy limit with small momentum transfer.

For the purposes of this paper, we assume that the $2 \rightarrow 2$ eikonal amplitude of interest involves one incoming scalar (massive or massless) and is mediated by graviton exchanges; the other particle could be any particle available in quantum field theory of mass greater than or equal to the scalar. The key is to interpret the latter as the source for a curved space-time, just like the ultraboosted scalar is the source for a shockwave metric. For instance, a massive scalar would correspond to the Schwarzschild metric[3], while a massive

---

[1]See also [17, 41] for a recent account of the relations between eikonal exponentiation and the classical limit of scattering amplitudes.

[2]There is also an intriguing relationship between eikonal factorization and scattering near black hole event horizons [52–55]. These works observe eikonal exponentiation for $2 \rightarrow 2$ scattering in a black hole space-time at *small* distances (i.e., very close to the event horizon). This a different from our set-up, where we relate $2 \rightarrow 2$ amplitudes in flat space to $1 \rightarrow 1$ amplitudes at *large* distances in curved backgrounds and systematically connect the eikonal phase to the space-time curvature.

[3]A relation between massive scalar eikonal scattering and scattering in Schwarzschild was observed long ago [47], but in the quantum mechanical framework of potential scattering, which is not covariant or fully relativistic.

particle with infinite quantum spin would relate to a Kerr black hole, as established in a series of remarkable recent papers [35–38, 56–64].

Following 't Hooft, we propose that the $2 \to 2$ eikonal scattering amplitude between a scalar and this other 'source' particle is equal to $1 \to 1$ scalar scattering on the curved space-time defined by the source. We give a precise formula for this $1 \to 1$ scattering amplitude on *any* stationary space-time in terms of a boundary integral; to avoid potential strong-field effects (e.g., particle creation) which would spoil the existence of a S-matrix [65–69], this boundary integral is localized at large distances from the source, so that only the linearized gravitational field plays a role. We find that this $1 \to 1$ scattering amplitude, $M_2$, always has the structure of an eikonal amplitude:

$$M_2 = \frac{2\pi \, \delta(p^{0\prime} - p^0)}{4M} \, \mathcal{M}_{\text{eik}} \,, \tag{1.1}$$

where $p'^\mu$, $p^\mu$ are the incoming and outgoing momenta of the scalar of mass $m$, $M$ is the Arnowitt-Deser-Misner (ADM) mass of the background with momentum $P^\mu$ and the remaining part $\mathcal{M}_{\text{eik}}$ is to be identified as an eikonal amplitude (stripped of an overall delta function).

We evaluate this $1 \to 1$ amplitude in the background space-times of the Schwarzschild and Kerr black holes. In the Schwarzschild case, we recover the known results for gravitational scattering of massive scalars in the eikonal regime, which provides a consistency check for the proposal. In Kerr, we obtain an eikonal amplitude corresponding to the exponentiation of the tree-level $2 \to 2$ amplitude between a massive scalar and a massive particle of arbitrarily large quantum spin [35]. This result can be viewed as evidence in favor of eikonal exponentiation with spin, as well as an alternative derivation of the tree-level amplitude.

The paper is organized as follows: Section 2 provides a brief review of eikonal scattering in QFT and the structure of eikonal amplitudes, followed by a discussion of $1 \to 1$ scalar scattering amplitudes in curved background space-times. As a warm-up, Section 3 reviews 't Hooft's calculation linking $1 \to 1$ massless scalar scattering on a shockwave space-time to the ultra-relativistic limit of eikonal scalar scattering. We then go on to show how to evaluate the $1 \to 1$ scattering amplitude for scalars in any stationary background space-time in Section 4, using a large-distance limit of the linearized background to ensure a well-defined S-matrix. Section 5 computes the amplitude for a Schwarzschild background, including a detailed analysis of its saddle points, poles and zeros. In Section 6, we compute the amplitude in a Kerr background, where we find that the resulting eikonal amplitude exhibits a factorization akin to Kawai-Lewellen-Tye factorization in string theory [70]. There is also a rich structure to the poles and zeros of the amplitude, which is explored. Section 7 concludes, while Appendix A includes some technical details on confluent hypergeometric functions.

We work in the mostly negative signature $(+, -, -, -)$ and follow notation from [71] where hats on integral measures and delta functions denote factors of $2\pi$: $\hat{\mathrm{d}}^n\omega := \mathrm{d}^n\omega/(2\pi)^n$ and $\hat{\delta}^n(k) := (2\pi)^n \, \delta^n(k)$.

## 2 Eikonal amplitudes from scattering in curved space-time

For our purposes, *eikonal scattering* will refer to $2 \to 2$ scattering of gravitationally-coupled particles in the high-energy regime with small momentum transfer (i.e., $-t \ll s$ in terms of the Mandelstam variables). The leading dominant diagrams in this limit are ladders and crossed ladders whose rungs are exchanges of the gravitons [9]. We say that *eikonal exponentiation* holds when this infinite series of ladder diagrams can be resummed into an *eikonal amplitude* (cf., [3]).

This eikonal amplitude is controlled by the *eikonal phase* $\chi$, which at leading order in the eikonal expansion is given by the inverse transverse Fourier transform of the single exchange diagram $A_4$ (or Born amplitude). Assuming two particles with incoming momenta $p_1^\mu$, $p_2^\mu$ and outgoing momenta $p_1^{\mu\prime}$, $p_2^{\mu\prime}$ then the leading order eikonal phase and associated eikonal amplitude can be written in a covariant form as:

$$\chi_1(x_\perp) := \hbar \int \hat{\mathrm{d}}^4 q \, \hat{\delta}(2p_1 \cdot q) \, \hat{\delta}(2p_2 \cdot q) \, \mathrm{e}^{\mathrm{i}\, q \cdot x/\hbar} \, A_4(q) \, , \tag{2.1}$$

$$\mathrm{i}\mathcal{M}_{\mathrm{eik}}(q_\perp) = \frac{4\sqrt{(p_1 \cdot p_2)^2 - m_1^2 m_2^2}}{\hbar^2} \int \mathrm{d}^2 x_\perp \, \mathrm{e}^{-\mathrm{i}\, q_\perp \cdot x_\perp / \hbar} \left( \mathrm{e}^{\mathrm{i}\chi_1(x_\perp)/\hbar} - 1 \right) \, , \tag{2.2}$$

where the two-dimensional impact parameter $x_\perp$ is orthogonal to $p_1^\mu$ and $p_2^\mu$ and conjugate to the (small) momentum exchange $q^\mu := p_1^\mu - p_1^{\prime\,\mu}$. Given $\chi_1$, one may further perform the $x_\perp$ integrals to obtain a closed form expression for the eikonal amplitude.

A well-studied example of eikonal exponentiation is the case of gravitationally coupled massive scalars. For equal masses, the leading order eikonal phase is [13, 47]

$$\chi_1(x_\perp) = \pi G \, \frac{(s - 2m^2)^2 - 2m^4}{\sqrt{s(s - 4m^2)}} \int \hat{\mathrm{d}}^2 \ell \, \frac{\mathrm{e}^{\mathrm{i}\, \ell \cdot x_\perp}}{\ell^2 + \mu^2 - \mathrm{i}\epsilon} \, , \tag{2.3}$$

where $G$ is Newton's constant, $s = (p_1 + p_2)^2$, $\ell$ is a 2-dimensional vector with units of an inverse length and the arbitrary scale $\mu$ serves to regulate infrared (IR) divergences. The impact parameter integral in the eikonal amplitude can be evaluated to give [47]

$$\mathrm{i}\,\mathcal{M}_{\mathrm{eik}}(q_\perp) = \frac{2\pi}{\mu^2 \hbar^2} \sqrt{s(s - 4m^2)} \frac{\Gamma(1 - \mathrm{i}\,\alpha(s))}{\Gamma(\mathrm{i}\alpha(s))} \left( \frac{4\hbar^2 \mu^2}{q_\perp^2} \right)^{1 - \mathrm{i}\alpha(s)} \, , \tag{2.4}$$

for

$$\alpha(s) := G \, \frac{(s - 2m^2)^2 - 2m^4}{\sqrt{s(s - 4m^2)}} \, . \tag{2.5}$$

It is easy to see that this eikonal amplitude is simply the tree-level/Born amplitude with single graviton exchange times a phase, $\mathcal{M}_{\mathrm{eik}} \sim A_4 \mathrm{e}^{\mathrm{i}\varphi}$, with the phase containing all dependence on the IR regulator.

Similarly, the ultra-relativistic limit of this scattering process corresponds to the regime where the scalar masses becomes negligible. In this case, the particles follow lightlike trajectories so it is natural to work in lightfront coordinates where $p_{i=1,2}^\mu = (p_{+\,i}, p_{-\,i}, p_{\perp\,i})$,

and so forth. In this case, we have $s = p_{+1} p_{-2}$ and the eikonal phase and amplitude simplify to

$$\chi_1(x_\perp) = \pi\, G\, s \int \hat{\mathrm{d}}^2\ell \, \frac{\mathrm{e}^{\mathrm{i}\,\ell_\perp \cdot x_\perp}}{\ell^2 + \mu^2 - \mathrm{i}\epsilon}\,, \tag{2.6}$$

where $\mu$ is an infra-red regulator, and the amplitude reads after Fourier transform to momentum space:

$$\mathrm{i}\,\mathcal{M}_{\mathrm{eik}}(q_\perp) = \frac{2\,\pi\,s}{\hbar^2 \mu^2}\, \frac{\Gamma(1 - \mathrm{i}\,G\,s)}{\Gamma(\mathrm{i}\,G\,s)}\, \left(\frac{4\hbar^2 \mu^2}{q_\perp^2}\right)^{1 - \mathrm{i}\,G\,s}\,, \tag{2.7}$$

respectively. Once again, this takes the form of the single exchange Born amplitude multiplied by a phase. This is seen by making use of the Gamma function identity $\Gamma(1 + x) = x\Gamma(x)$ which produces an extra factor of $Gs$ in the numerator, giving the usual $s^2/t$ behaviour of a single graviton exchange (since $t = q_\perp^2$).

Over thirty years ago, 't Hooft gave an alternative derivation of the eikonal amplitude for ultra-relativistic, gravitationally-coupled scalars which explains both eikonal exponentiation and the classicality of the eikonal amplitude [9]. By taking one of the ultraboosted scalars to source a gravitational shockwave [42, 43], the $2 \to 2$ scattering process is recast as semi-classical $1 \to 1$ scattering of a massless scalar in the shockwave space-time. By solving the wave equation in the shockwave background, 't Hooft was able to compute this $1 \to 1$ scattering amplitude and showed that it is indeed equal (up to an overall normalization) to the ultra-relativistic eikonal amplitude (2.7) for scalars.

However, the essential statement underlying 't Hooft's original result is potentially *much* more general than this ultra-relativistic scalar example. Indeed, it is natural to propose a generic correspondence between gravitational eikonal $2 \to 2$ scattering on the one hand, and $1 \to 1$ scattering on a classical, curved space-time whose source corresponds to one of the scattering states in the eikonal picture. Of course, some care is required for this generalised interpretation, since the curved space-time must be chosen in such a way that the $1 \to 1$ scattering process is well-defined.

Let us assume that the eikonal amplitude of interest involves one incoming scalar and is mediated by graviton exchanges; the other incoming particle could be another scalar, or something else (e.g., a spinning particle). Suppose this other particle has a natural interpretation in terms of a curved space-time: for instance, a massless scalar would correspond to the shockwave metric. Thus, we wish to calculate the 2-point, or $1 \to 1$, amplitude of a scalar in this fixed, non-perturbative space-time. Following the 'perturbiner' approach to scattering amplitudes (cf., [72–77]), this corresponds to evaluating the quadratic part of the gravitationally-coupled scalar action on-shell.

For a complex scalar field of mass $m$ in a curved background space-time $(M, g)$, one must therefore consider

$$S[\Phi] = \int_M \mathrm{d}^4 x \, \sqrt{|g|}\, \left(g^{\mu\nu}\, \partial_\mu \Phi(x)\, \partial_\nu \bar{\Phi}(x) - \frac{m^2}{\hbar^2}\, |\Phi|^2(x)\right)\,, \tag{2.8}$$

where $g_{\mu\nu}$ is the space-time metric and $|g|$ is the absolute value of its determinant. Here, the metric $g_{\mu\nu}$ is treated as fixed and non-dynamical. In other words, we are working within the

framework of background field theory (cf., [78–82]), where the scalar field is fully dynamical but the metric is treated as a fixed, non-perturbative, classical background.

We can express this action as a boundary term by evaluating (2.8) on-shell on solutions to the free equation of motion

$$\left( \Delta_g + \frac{m^2}{\hbar^2} \right) \Phi(x) = 0 \,, \tag{2.9}$$

where

$$\Delta_g \Phi(x) := |g|^{-1/2} \, \partial_\mu \left( |g|^{1/2} \, g^{\mu\nu} \, \partial_\nu \Phi(x) \right) \,, \tag{2.10}$$

stands for the action of the Laplacian of the background space-time. It is straightforward to integrate-by-parts to re-write the free action as

$$S[\Phi] = - \int_M \mathrm{d}^4 x \, \sqrt{|g|} \, \bar{\Phi}(x) \left( \Delta_g + \frac{m^2}{\hbar^2} \right) \Phi(x) + \int_{\partial M} \mathrm{d}^3 y \, \sqrt{|h|} \, \bar{\Phi}(y, \bar{x}) \, n^\mu \nabla_\mu \Phi(y, \bar{x}) \,, \tag{2.11}$$

where we work in a local coordinate system $x^\mu = (y^i, \bar{x})$ on $M$ for which the boundary $\partial M$ is given by $\bar{x} = \text{constant}$, $h_{ij}$ is the induced metric on this boundary and $n^\mu$ a normal vector to the boundary. Here, we intend for this 'boundary' $\partial M$ to include infinite regions (i.e., regions which would correspond to finite boundaries under conformal compactification).

Evaluated on-shell, the first term in (2.11) vanishes by virtue of the background-coupled free equation of motion. Therefore, we are left with

$$S[\Phi] = \int_{\partial M} \mathrm{d}^3 y \, \sqrt{|h|} \, \bar{\Phi}(y, \bar{x}) \, n^\mu \nabla_\mu \Phi(y, \bar{x}) \,, \tag{2.12}$$

Following the pertubiner approach, define the following object:

$$\Phi^{[2]}(x) := \epsilon_1 \phi_{\mathrm{in}}(x) + \epsilon_2 \phi_{\mathrm{out}}(x) \,. \tag{2.13}$$

The $\{\epsilon_i\}$ are complex parameters that will eventually be thought of as infinitesimal; $\phi_{\mathrm{in}}(x)$ is a solution to (2.9) in absence of gravity while $\phi_{\mathrm{out}}(x)$ when a gravitational field is present. Specifying the asymptotic behaviour of these solutions is equivalent to applying an LSZ reduction formula as it specifies whether $\phi_{\mathrm{in}}(x)$ and $\phi_{\mathrm{out}}(x)$ looks like an 'in' or 'out' state.

We can now define a 2-point tree-level scattering amplitude in a curved background as the multi-linear piece of the classical action:

$$M_2 := \frac{1}{\hbar^2} \left. \frac{\partial^2 S \left[ \Phi^{[2]} \right]}{\partial \bar{\epsilon}_1 \partial \epsilon_2} \right|_{\epsilon_1 = \epsilon_2 = 0} \,. \tag{2.14}$$

On a flat background, this 2-point function is clearly vanishing, but on a curved space-time it can be expressed as a non-vanishing boundary term

$$M_2 = \frac{1}{\hbar^2} \int_{\partial M} \mathrm{d}^3 y \, \sqrt{|h|} \, \bar{\phi}_{\mathrm{in}}(y, \bar{x}) \, n^\mu \nabla_\mu \phi_{\mathrm{out}}(y, \bar{x}) \,. \tag{2.15}$$

The central idea of this paper is that when the source of the background space-time admits a particle-like interpretation, $M_2$ is equal (up to normalization factors and a energy-conserving delta function) to the eikonal amplitude for scattering between this particle-like source and a scalar.

Of course, there are some subtleties associated with this interpretation of formula (2.15). In the first instance, to truly interpret $M_2$ as a scattering amplitude one must assume that the background space-time admits an S-matrix, in the sense that there are asymptotically flat in- and out-regions and there is no particle creation between them (cf., [65–69]). However, when the space-time is asymptotically flat but has particle creation (e.g., any Kerr-Newman black hole) one can still treat (2.15) as a scattering amplitude provided the incoming and outgoing states only probe far-field regions where particle creation can be neglected.

## 3   Warm-Up: Ultrarelativistic eikonal from shockwave backgrounds

Let us first review the most well-studied version of the correspondence between eikonal and curved-background scattering: the ultrarelativistic limit of eikonal scalar scattering, where the scalar masses are negligible. Consider the shockwave metric [42, 43]:

$$\mathrm{d}s^2 = 2\,\mathrm{d}x^-\,\mathrm{d}x^+ - (\mathrm{d}x^\perp)^2 + 4G\,P_-\,\delta(x^-)\,\log(\mu\,|x^\perp|)\,(\mathrm{d}x^-)^2\,, \tag{3.1}$$

where $\mu$ is an arbitrary mass scale. The quantity $P_-$ can be characterized by feeding this metric into the Einstein equations to give the stress tensor

$$T_{\mu\nu} = P_-\,\delta_\mu^-\,\delta_\nu^-\,\delta(x^-)\,\delta^2(x^\perp)\,, \tag{3.2}$$

which is that of a source of energy $P_-$ moving on the lightfront $x^- = 0$ and localized at the origin in the transverse plane. In other words, the shockwave is sourced by an ultraboosted particle, which is exactly what we want for the ultrarelativistic limit of eikonal scalar scattering.

Now we wish to compute the $1 \to 1$ classical scattering amplitude for a scalar in the shockwave space-time. With a free incoming wave $\phi_{\mathrm{in}} = \mathrm{e}^{-\mathrm{i}\,p\cdot x/\hbar}$, the outgoing, scattered, wave is determined by solving the scalar wave equation at general lightfront time in the shockwave metric. Since the shockwave is flat before and after the $x^- = 0$ lightfront, this is achieved by a patching argument on the two Minkowski regions [9, 47, 83–85]:

$$\phi_{\mathrm{out}}(x) = \Theta(-x^-)\,\mathrm{e}^{-\mathrm{i}\,p'\cdot x/\hbar} + \Theta(x^-)\frac{1}{\hbar^2}\int \hat{\mathrm{d}}^2\ell_\perp\,W(p'-\ell)\,\mathrm{e}^{-\mathrm{i}\ell\cdot x/\hbar}\Big|_{\ell_+ = p'_+}\,, \tag{3.3}$$

where $\Theta(x^-)$ is the Heaviside step function and the transverse momentum integrals are weighted by

$$W(\ell) := \int \mathrm{d}^2 y^\perp\,\mathrm{e}^{-\mathrm{i}\,\ell_\perp\cdot y^\perp/\hbar}\,\mathrm{e}^{-4\mathrm{i}\,G\,\ell_+\,P_-\,\log(\mu\,|y^\perp|)/\hbar}\,, \tag{3.4}$$

where the dependence on the shock profile itself enters. This choice of outgoing wave ensures an appropriate LSZ truncation, so that no scattering occurs before the $x^- = 0$ lightfront. Using the Klein-Gordon inner product [86], one can verify that there is no particle creation; combined with asymptotically flat in- and out-regions, this means that the space-time admits a semi-classical S-matrix (cf., [9, 84, 87, 88]).

When evaluating (2.15) with these choices of $\phi_{\mathrm{in}}$ and $\phi_{\mathrm{out}}$, there are in principle four distinct boundary contributions: those at past/future infinity ($x^- = \pm\infty$) and on either

side of the shockwave lightfront ($x^- = 0^\pm$), depicted in fig. 1. The contributions at infinity will vanish since both asymptotic regions are diffeomorphic to Minkowski space, and the i$\epsilon$-prescription is implicitly in play to ensure that the wavefunctions are asymptotically flat.

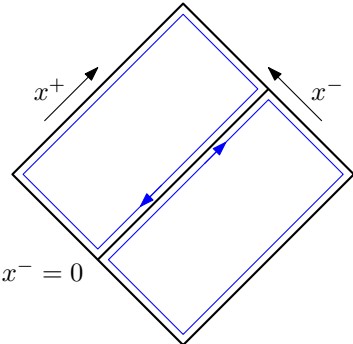

**Figure 1**. Boundary regions on a shockwave-background

Thus, the 2-point amplitude is entirely localized on both sides of the $x^- = 0$ lightfront:

$$M_2 = \frac{1}{\hbar^2} \lim_{\epsilon \to 0} \left( \int \mathrm{d}x^+ \, \mathrm{d}^2 x^\perp \, \bar{\phi}_\mathrm{in} \, \partial_+ \phi_\mathrm{out}|_{x^- = \epsilon} - \int \mathrm{d}x^+ \, \mathrm{d}^2 x^\perp \, \bar{\phi}_\mathrm{in} \, \partial_+ \phi_\mathrm{out}|_{x^- = -\epsilon} \right) . \tag{3.5}$$

The second term can be evaluated immediately, since there is no scattering before $x^- = 0$:

$$\frac{1}{\hbar^2} \lim_{\epsilon \to 0} \int \mathrm{d}x^+ \, \mathrm{d}^2 x^\perp \, \bar{\phi}_\mathrm{in} \, \partial_+ \phi_\mathrm{out}|_{x^- = -\epsilon} = -\mathrm{i} \, p_+ \, \hat{\delta}^3_{+,\perp}(p - p') . \tag{3.6}$$

This accounts for the (subtraction of the) forward scattering contribution to the amplitude. The first term, on the other hand, is given by:

$$\frac{1}{\hbar^2} \lim_{\epsilon \to 0} \int \mathrm{d}x^+ \, \mathrm{d}^2 x^\perp \, \bar{\phi}_\mathrm{in} \, \partial_+ \phi_\mathrm{out}|_{x^- = \epsilon} =$$
$$- \mathrm{i} \, p_+ \, \hat{\delta}(p_+ - p'_+) \frac{1}{\hbar^4} \lim_{\epsilon \to 0} \int \mathrm{d}^2 x^\perp \, \hat{\mathrm{d}}^2 \ell_\perp \, W(p' - \ell_\perp) \, \mathrm{e}^{-\mathrm{i}\,(\ell - p)_\perp \cdot x^\perp / \hbar + \mathrm{i}\, p'_- \epsilon / \hbar}$$
$$= -\frac{\mathrm{i} \, p_+}{\hbar^2} \, \hat{\delta}(p_+ - p'_+) \, W(q_\perp) , \tag{3.7}$$

where $q_\perp := p_\perp - p'_\perp$ as usual. Putting all of this together, we find

$$\begin{aligned} M_2 &= \frac{\mathrm{i} \, p_+}{\hbar^2} \, \hat{\delta}(p_+ - p'_+) \left( W(q_\perp) - \hbar^2 \hat{\delta}^2(q_\perp) \right) \\ &= \frac{\mathrm{i} \, p_+}{\hbar^2} \, \hat{\delta}(p_+ - p'_+) \int \mathrm{d}^2 x^\perp \, \mathrm{e}^{-\mathrm{i}\, q_\perp \cdot x^\perp / \hbar} \left( \mathrm{e}^{-4\mathrm{i}\, G \, p_+ \, P_- \, \log(\mu \, |x^\perp|) / \hbar} - 1 \right) , \end{aligned} \tag{3.8}$$

where we used the definition of $W$ to get the second line.

This is equal to the ultrarelativistic eikonal amplitude (2.7) up to overall factors:

$$M_2 := \frac{\hat{\delta}(p_+ - p'_+)}{8 P_-} \, \mathcal{M}_\mathrm{eik}(q_\perp) , \tag{3.9}$$

Thus, we recover the original observation of 't Hooft: the phase shift of a massless scalar wavefunction crossing a shockwave background contains the leading eikonal resummation for gravitational scattering of massless scalars [9].

## 4 Eikonal amplitude from stationary backgrounds

In the case of the shockwave, relevant for ultrarelativistic scalar eikonal scattering, the background space-time admits an S-matrix and it is possible to solve for the outgoing scattered wave exactly. But suppose we want to compute the $1 \to 1$ amplitude on more general backgrounds, which don't admit an S-matrix or for which it is difficult to solve the wave equation exactly, such as a black hole?

Here, we present a covariant framework to obtain an appropriate classical $1 \to 1$ scattering amplitude of a massive scalar on any stationary background space-time. There are two key steps: the first is to consider the large-distance (i.e., far from the source) regime of the linearized background to ensure a well-defined S-matrix, and then to solve the wave equation perturbatively with a WKB ansatz and appropriate boundary conditions.

### 4.1 Asymptotic states on curved space-times

For the specific case of a shockwave background, we have seen that the notion of outgoing state is in correspondence with a solution to the wave equation on (3.1). For generic backgrounds, in order to define a $1 \to 1$ amplitude, one has to solve the wave equation on a curved background to define a proper outgoing state. To achieve this for the general setting of weakly curved, stationary space-times, we focus on a linearized gravitational field assuming only the existence of a time-like Killing vector $\partial_t$:

$$\mathrm{d}s^2 = \eta_{\mu\nu}\,\mathrm{d}x^\mu\,\mathrm{d}x^\nu + h_{\mu\nu}(x)\,\mathrm{d}x^\mu\,\mathrm{d}x^\nu \,. \tag{4.1}$$

On this background, the wave equation becomes

$$\left(\Box + \frac{m^2}{\hbar^2}\right)\phi(x) = J_{\mathrm{eff}}(x) \,, \tag{4.2}$$

where

$$J_{\mathrm{eff}}(x) := h^{\mu\nu}(x)\partial_\mu\partial_\nu\phi(x) \,, \tag{4.3}$$

defines the self-interaction of the field as an effective source $J_{\mathrm{eff}}(x)$, in analogy with the standard method of solving (4.2) in presence of matter. To describe an outgoing state, we look for solutions to (4.2) which can be written as a sum of an incoming free wave $\phi_{\mathrm{in}}(x)$ and an outgoing distorted wave with a $1/r$ fall-off typical of scattering processes. Following [89], we can write these solutions in the asymptotic region as

$$\phi(x) = \phi_{\mathrm{in}}(x) + \frac{1}{4\pi r}\int_0^\infty \hat{\mathrm{d}}\omega \ \tilde{J}_{\mathrm{eff}}(\hbar\omega, \hbar\omega\hat{\mathbf{n}})\,\mathrm{e}^{-\mathrm{i}\,t\omega+\mathrm{i}\,\omega r} \,, \tag{4.4}$$

where the distorted outgoing wave depends on the effective source in momentum space

$$\tilde{J}_{\mathrm{eff}}(k) := \int \mathrm{d}^4x \ J_{\mathrm{eff}}(x)\,\mathrm{e}^{\mathrm{i}\,k\cdot x/\hbar}\Big|_{k^\mu=\hbar(\omega,\omega\hat{\mathbf{n}})} \,, \tag{4.5}$$

and $\hat{\mathbf{n}}$ is a 3-dimensional unit vector on the celestial sphere.

At this point, the solution to (4.2) is still implicit as the source in (4.4) is a function of the field itself. A perturbative ansatz to evaluate (4.5) is provided by evaluating the effective source on an eikonal solution for the outgoing wave

$$\tilde{J}_{\text{eff}}(k) = \int \mathrm{d}^4x \; h^{\mu\nu}(x)\, \partial_\mu \partial_\nu \phi_{\text{eik}}(x)\, \mathrm{e}^{\mathrm{i}\,k\cdot x/\hbar}\Big|_{k^\mu = \hbar(\omega,\omega\hat{\mathbf{n}})}\,, \qquad (4.6)$$

where by definition $\phi_{\text{eik}}(x)$ is solution to equation (4.2), for $\hbar \to 0$, of the form

$$\phi_{\text{eik}}(x) = \mathrm{e}^{\mathrm{i}\,\chi(x)/\hbar}\,, \qquad \chi(x) = \sum_{n=0}^{\infty} \chi_n(x)\,, \quad \chi_n(x) \sim G^n\,. \qquad (4.7)$$

One advantage in using the eikonal limit is that we can trade (4.2) for a simpler set of differential equations for $\chi(x)$. At leading and next-to-leading order in the gravitational coupling, these are

$$\begin{aligned}
\partial_\mu \chi_0(x)\, \partial^\mu \chi_0(x) &= m^2\,,\\
2\, \partial^\mu \chi_0(x)\, \partial_\mu \chi_1(x) &= h^{\mu\nu}(x)\, \partial_\mu \chi_0(x)\, \partial_\nu \chi_0(x)\,.
\end{aligned} \qquad (4.8)$$

The first of these is trivially solved by $\chi_0(x) = -p \cdot x$, corresponding to an incoming wave with on-shell momentum $p^\mu$. To provide a concrete example, we assume that this momentum is directed along the $z-$axis such that $p^\mu = (\sqrt{p_z^2 + m^2}, 0, 0, p_z)$. Turning to the second equation in (4.8), we assume that $\chi_1(x)$ is time-independent. This is always possible since the metric admits a time-like Killing vector, by assumption. Further imposing the boundary condition that one should have a trivial free field at $z \to -\infty$, we obtain

$$\begin{cases} 2\,p_z\, \partial_z \chi_1(x_\perp, z) = h^{\mu\nu}(x_\perp, z)\, p_\mu\, p_\nu\,,\\ \chi_1(x_\perp, z = -\infty) = 0\,, \end{cases} \qquad (4.9)$$

where $x_\perp$ denotes generic coordinates on the orthogonal plane to $z$.

These boundary conditions single out a unique solution given by

$$\chi_1(x_\perp, z) = \frac{1}{2p_z} \int_{-\infty}^{z} \mathrm{d}z'\, h^{\mu\nu}(x_\perp, z')\, p_\mu\, p_\nu\,. \qquad (4.10)$$

Thus, the desired wave in the eikonal limit is

$$\phi_{\text{eik}}(x) = \mathrm{e}^{-\mathrm{i}\,p\cdot x/\hbar + \mathrm{i}\,\chi_1(x_\perp, z)/\hbar}\,, \qquad (4.11)$$

Feeding this expression into (4.6), one reads off the effective source:

$$\tilde{J}_{\text{eff}}(k) = -\frac{2\,p_z}{\hbar^2} \int \mathrm{d}^2x_\perp\, \mathrm{d}z\, \mathrm{d}t\, \partial_z \chi_1(x_\perp, z)\; \mathrm{e}^{\mathrm{i}\,(k-p)/\hbar \cdot x + \mathrm{i}\,\chi_1(x_\perp, z)/\hbar}\Big|_{k^\mu = \hbar(\omega,\omega\hat{\mathbf{n}})}\,. \qquad (4.12)$$

This can be further simplified by performing the $t$-integration, to give

$$\tilde{J}_{\text{eff}}(k^0, k_\perp, k_z) = -\frac{2\,p_z}{\hbar}\, \hat{\delta}(p^0 - k^0) \int \mathrm{d}^2x_\perp\, \mathrm{d}z\, \partial_z \chi_1(x_\perp, z)\, \mathrm{e}^{-\mathrm{i}k_\perp \cdot x_\perp/\hbar - \mathrm{i}\,(k_z - p_z)\,z/\hbar + \mathrm{i}\,\chi_1(x_\perp, z)/\hbar}\,, \qquad (4.13)$$

where we identify $k_\perp := \hbar\omega\hat{\mathbf{n}}_\perp$, $k_z := \hbar\omega\mathbf{n}_z$ and $k^0 := \hbar\sqrt{\omega^2 + m^2}$.

By further making the small angle approximation, for which $|k_\perp| \ll k_z$ and $k_z \sim p_z$, one obtains a closed form for the outgoing wave with the desired boundary conditions:

$$\phi(x) = \phi_{\text{in}}(x) + \mathrm{i}\, p_z \, \frac{\mathrm{e}^{-\mathrm{i}\,t\,\sqrt{p_z^2+m^2}/\hbar + \mathrm{i}\,p_z\,r/\hbar}}{2\pi r\hbar} \int \mathrm{d}^2 x_\perp\, \mathrm{e}^{-\mathrm{i}\,q_\perp \cdot x_\perp/\hbar} \left( \mathrm{e}^{\mathrm{i}\,\chi_1(x_\perp, z=+\infty)/\hbar} - 1 \right), \quad (4.14)$$

where $q_\perp := p_z \hat{\mathbf{n}}_\perp$ This expression provides the notion of outgoing state in *any* linearized stationary background: specifying a given linearized metric uniquely fixes the outgoing wave. We can now proceed to see how these waves can be used to evaluate the $1 \to 1$ amplitude on any linearized stationary background.

## 4.2 Exponentiation from spatial infinity

In section 2, we saw that the $1 \to 1$ amplitude on a generic background can be expressed as a boundary term:

$$M_2 = \frac{1}{\hbar^2} \int_{\partial M} \mathrm{d}^3 y \, \sqrt{|h|(y)} \bar{\phi}_{\text{in}}(y, \bar{x})\, n \cdot \partial\phi_{\text{out}}(y, \bar{x}) \ , \quad (4.15)$$

where $\bar{x}$ denotes the variable on $M$ which specifies the boundary, while $y$ are coordinates on the boundary and $n$ a unit normal vector orthogonal to to the boundary. Now, suppose our background is stationary and admits spherical coordinates[4] $(t, r, \theta, \varphi)$, for which the metric is flat at spatial infinity where $r \to \infty$. To ensure that (4.15) makes sense as a scattering amplitude, we only consider contributions to (4.15) from the 'boundary' at spatial infinity; in other words, we consider only scattering at sufficiently large distances from any source.

To this end, we evaluate (4.15) by choosing $\bar{x} = r$ and taking $r = \infty$ while keeping the other variables $(t, \theta, \varphi)$ fixed. Within this limit, the determinant of the induced metric is trivial, leaving

$$M_2 = -\frac{1}{\hbar^2} \lim_{r \to \infty} \int_{S^2 \times \mathbb{R}} \mathrm{d}\theta\, \mathrm{d}\varphi\, \mathrm{d}t\, r^2 \sin(\theta)\, \bar{\phi}_{\text{in}}(t, r, \theta, \varphi)\, \partial_r \phi_{\text{out}}(t, r, \theta, \phi) \ . \quad (4.16)$$

We now specify the incoming and outgoing states: as incoming state we choose a plane wave with on-shell momentum $p^{\mu'}$ while for an outgoing state we choose a spherical wave in the small angle approximation with on-shell momentum $p^\mu$

$$\phi_{\text{out}}(x) = \frac{\mathrm{e}^{-\mathrm{i}\,tp^0/\hbar + \mathrm{i}\,p_z r/\hbar}}{r} f_p(\hat{\mathbf{n}}_\perp) \ , \quad (4.17)$$

where $f_p$ is defined by

$$f_p(\hat{\mathbf{n}}_\perp) := \frac{\mathrm{i}\,p_z}{2\pi\hbar} \int \mathrm{d}^2 x_\perp\, \mathrm{e}^{-\mathrm{i}\,k_\perp \cdot x_\perp/\hbar} \left( \mathrm{e}^{\mathrm{i}\,\chi_1(x_\perp)/\hbar} - 1 \right), \quad (4.18)$$

and we abbreviate $\chi_1(x_\perp) := \chi_1(x_\perp, z = +\infty)$.

---

[4]The space-time need only admit these coordinates locally, in a neighborhood of spatial infinity.

Feeding this into (4.16), and neglecting subleading terms in $1/r^2$, the $t$ integral can be performed immediately to give the energy conserving delta function expected for stationary backgrounds

$$M_2 = -\frac{\mathrm{i}p_z}{\hbar^2}\,\hat{\delta}(p^{0\prime} - p^0) \lim_{r\to\infty}\, r\, \mathrm{e}^{\mathrm{i}\,p_z\, r/\hbar} \int_0^\pi \mathrm{d}\theta\, \sin(\theta)\, \mathrm{e}^{-\mathrm{i}\,p_z'\, r\cos(\theta)/\hbar} \int_0^{2\pi} \mathrm{d}\phi\, f_p(\hat{\mathbf{n}}_\perp)\,. \quad (4.19)$$

A straightforward integration shows that the remaining integral is finite by including a proper i$\epsilon$-prescription at spatial infinity. In the small angle approximation

$$M_2 = -\frac{\mathrm{i}p_z\,\hat{\delta}(p^{0\prime} - p^0)}{\hbar^2} \int \mathrm{d}^2 x_\perp \mathrm{e}^{-\mathrm{i}q_\perp \cdot x_\perp/\hbar} \left( \mathrm{e}^{\mathrm{i}\chi_1(x_\perp)/\hbar} - 1 \right) \quad (4.20)$$

$$= \frac{p_z\,\hat{\delta}(p^{0\prime} - p^0)}{4\sqrt{(p\cdot P)^2 - m^2 M^2}}\, \mathcal{M}_{\mathrm{eik}}(q_\perp)\,. \quad (4.21)$$

This can be written in a more compact way as

$$M_2 = \frac{\hat{\delta}(p^{0\prime} - p^0)}{4M}\, \mathcal{M}_{\mathrm{eik}}(q_\perp)\,. \quad (4.22)$$

Up to normalization factors, the $1 \to 1$ amplitude we have computed has precisely the same structure of an eikonal amplitude (2.2), with $\chi_1$ playing the role of the eikonal phase. This structural equivalence holds for large-distance scattering in *any* stationary space-time, suggesting that 't Hooft's proposal linking eikonal and curved-background scattering should hold beyond the ultrarelativistic scalar case.

## 5    Schwarzschild background

Perhaps the most obvious non-trivial example of a stationary, asymptotically flat space-time is the Schwarzschild black hole. This is a non-spinning metric sourced by a massive, time-like worldline, so from the perspective of eikonal scattering should correspond to a massive scalar particle [13]. The connection between massive scalar eikonal scattering and scattering on Schwarzschild was first explored by Kabat and Ortiz [47], who translated the Schwarzschild problem into quantum mechanical Rutherford scattering. While it produces the correct eikonal amplitude, this method is not covariant and obscures the fully relativistic nature of the amplitude.

In this section, we use the method derived in the previous section to compute the eikonal phase and amplitude for $1 \to 1$ scattering of a massive scalar on the Schwarzschild space-time, finding the expected results for scalar eikonal scattering with arbitrary masses. We then analyse the saddle point and poles of the eikonal amplitude, which encode the Born amplitude and classical bound states of the system, respectively. While these have been studied previously in the literature [9, 47, 90], the details will prove instructive for later calculations in Section 6.

## 5.1 The eikonal phase for Schwarzschild

In equation (4.20), we derived a closed form for the $1 \to 1$ scalar scattering amplitude valid for the long-distance regime of any linearized stationary space-time. This amplitude was determined by the eikonal phase $\chi_1$ of the background, which is itself given by the linearized background metric in (4.10). This eikonal phase is easily rewritten in momentum space using the spatial Fourier transform for the metric

$$\tilde{h}^{\mu\nu}(q_\perp, q_z) := \int \mathrm{d}^2 x_\perp \, \mathrm{d}z \, \mathrm{e}^{-\mathrm{i}\, q_\perp \cdot x_\perp / \hbar - \mathrm{i} q_z z / \hbar} \, h^{\mu\nu}(x_\perp, z) \ . \tag{5.1}$$

With this, the eikonal phase becomes

$$\chi_1(x_\perp) = \frac{1}{2p_z} \int \hat{\mathrm{d}}^2 q_\perp \, \mathrm{e}^{-\mathrm{i}\, q_\perp \cdot x_\perp / \hbar} \, \tilde{h}^{\mu\nu}(q_\perp, q_z = 0) \, p_\mu \, p_\nu \ . \tag{5.2}$$

In a more covariant way, this can be expressed as

$$\chi_1(x_\perp) = 2M \int \hat{\mathrm{d}}^4 q \, \hat{\delta}(2P \cdot q) \, \hat{\delta}(2p \cdot q) \, \mathrm{e}^{-\mathrm{i}\, q_\perp \cdot x_\perp / \hbar} \, \tilde{h}^{\mu\nu}(q) \, p_\mu \, p_\nu \ , \tag{5.3}$$

where $P^\mu = (M, 0, 0, 0)$ and $p^\mu$ is the on-shell momenta of the probe. In this form, equation (5.3) is manifestly covariant and it can be used to infer the $1 \to 1$ amplitude when the source is no longer static by applying a Lorentz transformation: in this case, $x_\perp$ and $q_\perp$ will be on the orthogonal plane of $p^\mu$ and $P^\mu$. Another advantage of (5.3) is that it makes manifest the gauge invariance of our amplitude. This is easily checked by noticing that under a linear diffeomorphism the eikonal phase remains unchanged

$$\Delta\chi_1(x_\perp) = 2M \int \hat{\mathrm{d}}^4 q \, \hat{\delta}(2P \cdot q) \, \hat{\delta}(2p \cdot q) \, \mathrm{e}^{-\mathrm{i}\, q_\perp \cdot x_\perp / \hbar} \, \tilde{\xi}(q) \cdot p \, q \cdot p = 0 \ , \tag{5.4}$$

where $\tilde{\xi}^\mu(q)$ is the Fourier transform of the vector field $\xi^\mu(x)$ generating the diffeomorphism.

Our proposal is that the Schwarzschild eikonal phase should correspond to the 'standard' eikonal phase of $2 \to 2$ gravitational scattering of massive scalars. If this is true, then the eikonal phase is the inverse Fourier transform of a tree-level $2 \to 2$ scattering amplitude, and we can use (5.3) to read off this 4-point function:

$$A_4(q) = 2M \, \tilde{h}^{\mu\nu}(q_\perp) \, p_\mu \, p_\nu \ . \tag{5.5}$$

Given that the eikonal phase is gauge invariant, we can choose any coordinate system for the metric tensor. Picking for example harmonic coordinates for Schwarzschild, its linearized form is

$$h^{\mu\nu}(x) = \mathcal{P}^{\mu\nu\alpha\beta} \left( u_\alpha \, u_\beta \, \frac{4M\,G}{r} \right) \ , \tag{5.6}$$

where we have introduced the projector $\mathcal{P}^{\mu\nu\alpha\beta} = \frac{1}{2}\eta^{\mu\alpha}\eta^{\nu\beta} + \frac{1}{2}\eta^{\mu\beta}\eta^{\nu\alpha} - \frac{1}{2}\eta^{\mu\nu}\eta^{\alpha\beta}$ and $Mu_\alpha = P_\alpha$. Using (5.1) one obtains

$$A_4(q) = \frac{16\pi\,G}{\hbar} \frac{[2(P \cdot p)^2 - m^2 M^2]}{q^2} \ , \tag{5.7}$$

which is indeed the leading in $\hbar$ 4-point function for a massive particle scattering at tree-level on a static source. Equation (5.7) is also valid for arbitrary mass ratios and for a moving background source. The fact that the amplitude in the probe limit constrains the analogue arbitrary mass ratio result at leading order is a well known fact also noticed in observables such as the scattering angle [91, 92].

## 5.2 The eikonal amplitude on a Schwarzschild background

Having seen the properties of the eikonal phase for Schwarzschild, we can now proceed to compute the associated $1 \to 1$ amplitude. To do so, we first evaluate $\chi_1(x_\perp)$ using equation (5.6); in momentum space and in the center of mass frame, one obtains the covariant result

$$\chi_1(x_\perp) = -\frac{2G\left(2\left(p \cdot P\right)^2 - m^2 M^2\right)}{\sqrt{(p \cdot P)^2 - m^2 M^2}}\, \log(\mu\,|x_\perp|)\ , \tag{5.8}$$

where $\mu$ is a mass scale introduced to regulate IR divergences.

At this point, the $1 \to 1$ amplitude for a scalar of mass $m$ on a Schwarzschild background of mass $M$ takes the form

$$\mathcal{M}_{\mathrm{eik}}(q_\perp) = -\frac{4\mathrm{i}\,\sqrt{(p \cdot P)^2 - m^2 M^2}}{\hbar^2}\left(\frac{I(q_\perp)}{\mu^{2\alpha(s)}} - \hat{\delta}^2(q_\perp)\right)\ , \tag{5.9}$$

where

$$I(q_\perp) := \int \mathrm{d}^2 x_\perp\, \mathrm{e}^{-\mathrm{i}\,q_\perp \cdot x_\perp/\hbar}\,|x_\perp|^{-2\mathrm{i}\,\alpha(s)}\quad . \tag{5.10}$$

and $\alpha(s)$ reads

$$\alpha(s) := \frac{G\left[(s - m^2 - M^2)^2 - 2\,m^2 M^2\right]}{\hbar\,\sqrt{s - (m+M)^2}\,\sqrt{s - (m - M)^2}} = \frac{GmM}{\hbar}\,\frac{\gamma(v)}{v}\left(1 + v^2\right)\ , \tag{5.11}$$

as a function of the center of mass energy Mandelstam invariant $s := (p + P)^2$, where the relativistic gamma factor also reads

$$\gamma(v) = (1 - v^2)^{-1/2} := \frac{p \cdot P}{m\,M} = \frac{s^2 - m^2 - M^2}{2m\,M}\ . \tag{5.12}$$

Neglecting the forward scattering contribution given by the delta function in (5.9), the non-trivial part of the amplitude is the 2-dimensional integral $I$, which is known explicitly (e.g., [47]):

$$I(q_\perp) = \pi\,\frac{\Gamma(1 - \mathrm{i}\,\alpha(s))}{\Gamma(\mathrm{i}\,\alpha(s))}\left(\frac{4\hbar^2}{q_\perp^2}\right)^{1 - \mathrm{i}\,\alpha(s)}\ . \tag{5.13}$$

While this Fourier transform is not difficult, in the next section we will encounter generalizations of this sort of integrals which will require an involved analytic continuation to evaluate, analogous to KLT factorization in string theory [70]. Thus, as a warm-up for these later calculations, we will now re-derive (5.13) from scratch using this factorization method.

Our starting point is (5.10), rewritten in terms of complex variables $z$ and $q$ defined by $x_\perp = (x, y) \to z = x + \mathrm{i}y$, $q_\perp = (q_x, q_y) \to q = q_x - \mathrm{i}q_y$, so that

$$I(q_\perp) = \int \mathrm{d}x\,\mathrm{d}y e^{-\mathrm{i}(zq + \bar{z}\bar{q})/2\hbar}\, z^{\alpha}\, \bar{z}^{\alpha}\,, \tag{5.14}$$

where we adopt a shorthand

$$\boldsymbol{\alpha} := -\mathrm{i}\alpha(s) \tag{5.15}$$

to abbreviate the $s$-dependence of the integrand and avoid carrying a sign and a factor of i everywhere. Our goal is to analytically continue the integration over $y$ to the imaginary axis $y \to \tilde{y} \propto \mathrm{i}y$, so that $z, \bar{z}$ become two independent real variables $u = x - \tilde{y}$ and $v = x + \tilde{y}$ and we can perform the $\int \mathrm{d}u$ and $\int \mathrm{d}v$ integrals independently.

This cannot be done by simply rotating the $y$-contour by $\pi/2$, as the integrand does not converge at both positive and negative values of $\Im(y)$ because of the exponential. Rather, we need to fold the $y$-contour in the upper or lower half-plane. Indeed, in terms of $x$ and $y$, the original integrand reads

$$f(x, y) = e^{-\mathrm{i}(xq_x + yq_y)/\hbar}(x + \mathrm{i}y)^{\boldsymbol{\alpha}}\, (x - \mathrm{i}y)^{\boldsymbol{\alpha}}\,, \tag{5.16}$$

so if we assume $q_y > 0$ (i.e., $\Im(q) < 0$), we need to continue the integral with $\Im(y) > 0$. Since this integrand has cuts starting at $\mathrm{i}|x|$ which extend to infinity, our contour must avoid them. The folding is then explicitly performed by considering the original integrand, integrated along the closed contour $\mathcal{C}_L$ of figure 2. If $q_y < 0$, we simply close the contour in the lower-half plane.

Since the integrand is holomorphic within $\mathcal{C}_L$, Cauchy's theorem relates the original integral along real $y$ to an integral along the imaginary axis given by the contour $\mathcal{C}_1 \cup \mathcal{C}_2$:

$$\oint_{\mathcal{C}_L} \mathrm{d}y\, f(x, y) = 0 \underset{L \to \infty}{\Longrightarrow} \int_{-\infty}^{\infty} \mathrm{d}y\, f(x, y) = \int_{\mathcal{C}_1 \cup \mathcal{C}_2} \mathrm{d}y\, f(x, y) \tag{5.17}$$

The variable $x$ is integrated over the whole real line, so let us first look at the case $x > 0$. The cut in the upper half plane is generated by the factor $(x + \mathrm{i}y)^\alpha$, which has a branch point at $y = \mathrm{i}x$. Along the vertical contours, where $y = \mathrm{i}\tilde{y}$ for $\tilde{y} > x$, $z$ and $\bar{z}$ become $z \to u = x - \tilde{y}$ and $\bar{z} \to v = x + \tilde{y}$. However, $x - \tilde{y}$ is negative, so $(x - \tilde{y})^{\boldsymbol{\alpha}}$ acquires a phase. On $\mathcal{C}_1$, $y = \mathrm{i}\tilde{y} - \epsilon$, thus $(x - \tilde{y} - \mathrm{i}\epsilon)^{\boldsymbol{\alpha}} = |\tilde{y} - x|^{\boldsymbol{\alpha}}\, e^{-\mathrm{i}\pi\boldsymbol{\alpha}}$, while on $\mathcal{C}_2$ we have $(x - \tilde{y} + \mathrm{i}\epsilon)^{\boldsymbol{\alpha}} = |\tilde{y} - x|^{\boldsymbol{\alpha}}\, e^{\mathrm{i}\pi\boldsymbol{\alpha}}$.

Overall, we obtain

$$\int_{\mathcal{C}_1 \cup \mathcal{C}_2} \mathrm{d}y\, f(x, y) = 2\mathrm{i}\,\sin(\pi\boldsymbol{\alpha}) \int_x^{\infty} \mathrm{i}\,\mathrm{d}\tilde{y}\, e^{-\mathrm{i}\,(uq + v\bar{q})/2\hbar}|u|^{\boldsymbol{\alpha}}\, v^{\boldsymbol{\alpha}}, \quad x > 0 \tag{5.18}$$

where the $\sin(\pi\boldsymbol{\alpha})$ comes from summing both phases $e^{\pm\mathrm{i}\pi\boldsymbol{\alpha}}$, with a sign from opposite orientations. In a slight abuse of notation, we use the $u, v$ variable as a shorthand in the integrand while we express the measure in terms of $x, \tilde{y}$.

For $x < 0$, nothing changes. The branch points at $\pm\mathrm{i}x$ exchange locations, but there is no phase associated to the superposed branches in the interval $[-|x|, |x|]$, and therefore

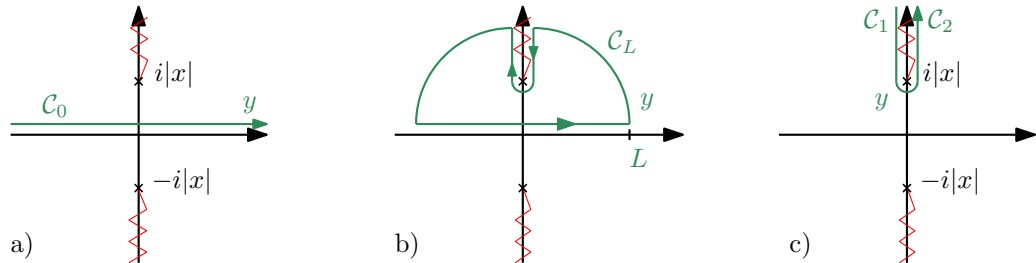

**Figure 2**. Contour deformation. a) Real contour for $y$ (green). b) Closed contour for which the arcs drop as $L \to \infty$ when $\Im(q) < 0$. c) The vanishing of the integral on $\mathcal{C}_L$ gives that $\int_{\mathcal{C}_0} = \int_{\mathcal{C}_1} + \int_{\mathcal{C}_2}$

the cut at $-ix$ still comes from the factor $(x + iy)^{\boldsymbol{\alpha}}$, hence the associated phase factor remains unchanged. This is depicted in fig. 3. We thus obtain the exact same phase factor as above, so that

$$\int_{\mathcal{C}_1 \cup \mathcal{C}_2} \mathrm{d}y\, f(x, y) = 2\mathrm{i}\,\sin(\pi\boldsymbol{\alpha}) \int_{-x}^{\infty} \mathrm{i}\,\mathrm{d}\tilde{y}\, \mathrm{e}^{-\mathrm{i}\,(uq+v\bar{q})/2\hbar}\, |u|^{\boldsymbol{\alpha}}\, v^{\boldsymbol{\alpha}}\,, \quad x < 0\,. \tag{5.19}$$

At this point, restoring the integral over $x$, we have expressed the original contour of integration as:

$$\int \mathrm{d}x\, \mathrm{d}y\, f(x, y) = -2\,\sin(\pi\boldsymbol{\alpha}) \left( \int_0^{\infty} \mathrm{d}x \int_x^{\infty} \mathrm{d}\tilde{y}\, \mathrm{e}^{-\mathrm{i}\,(uq+v\bar{q})/2\hbar}\, |u|^{\boldsymbol{\alpha}}\, v^{\boldsymbol{\alpha}} \right.$$
$$\left. + \int_{-\infty}^{0} \mathrm{d}x \int_{-x}^{\infty} \mathrm{d}\tilde{y}\, \mathrm{e}^{-\mathrm{i}\,(uq+v\bar{q})/2\hbar}\, |u|^{\boldsymbol{\alpha}}\, v^{\boldsymbol{\alpha}} \right)\,. \tag{5.20}$$

This integral is almost factorized in the $u$ and $v$ variables; it only remains to show that the $u$ and $v$ integration domains are independent. For $x > 0$, the integration domain is $\tilde{y} > x$, while for $x < 0$ we have $\tilde{y} > -x$. The union of these two domains, respectively colored in pale yellow and pale blue in fig. 4, piece up to the full $u < 0, v > 0$ quadrant, which achieves to prove the factorization.

Collecting all signs and factors, adding a $1/2$ for the Jacobian $\mathrm{d}x\mathrm{d}\tilde{y} = 1/2\mathrm{d}u\mathrm{d}v$, and using eq.(5.17), we find that

$$I(q_\perp) = -\sin(\pi\boldsymbol{\alpha}) \int_{-\infty}^{0} \mathrm{d}u \int_0^{\infty} \mathrm{d}v\, \mathrm{e}^{-\mathrm{i}\,(uq+v\bar{q})/2\hbar}\, |u|^{\boldsymbol{\alpha}}\, v^{\boldsymbol{\alpha}}\,. \tag{5.21}$$

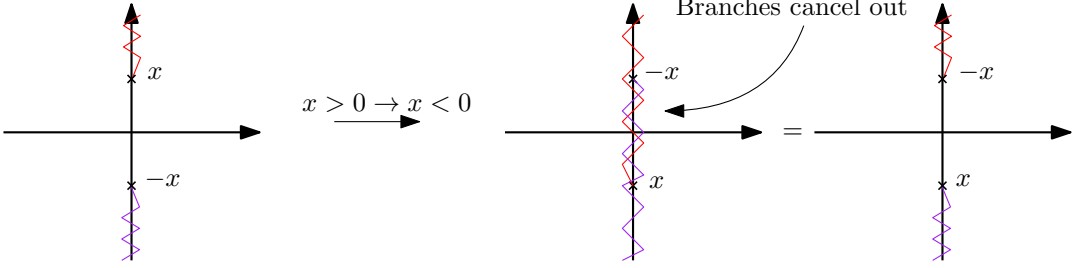

**Figure 3**. As cuts exchange location when $x$ becomes negative, the phase in the intermediate segment $[-|x|, |x|]$ vanishes and the upper cut still comes from $(x + iy)^{\boldsymbol{\alpha}}$.

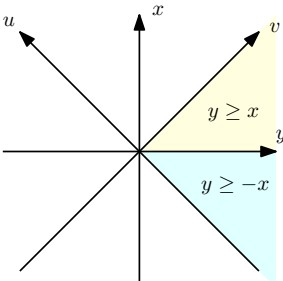

**Figure 4**. Quadrant of integration described in (5.20) and below.

Each integral can now be computed separately (remember $\Im(q) < 0$):

$$
\begin{aligned}
I_- &= \int_{-\infty}^{0} \mathrm{d}u \, \mathrm{e}^{-\mathrm{i}u\,q/2\hbar} \, |u|^{\boldsymbol{\alpha}} = \Gamma(1+\boldsymbol{\alpha}) \left(\frac{2\hbar}{\mathrm{i}q_\perp}\right)^{1+\boldsymbol{\alpha}} \\
I_+ &= \int_{0}^{\infty} \mathrm{d}v \, \mathrm{e}^{-\mathrm{i}v\,\bar{q}/2\hbar} \, v^{\boldsymbol{\alpha}} = \Gamma(1+\boldsymbol{\alpha}) \left(\frac{-2\hbar}{\mathrm{i}\bar{q}_\perp}\right)^{1+\boldsymbol{\alpha}}
\end{aligned}
\tag{5.22}
$$

for $\Re(\boldsymbol{\alpha}) > -1$. Overall, we obtain a KLT-like representation of the integral

$$
I(q_\perp) = -\sin(\pi\boldsymbol{\alpha}) \, I_- \, I_+ \,.
\tag{5.23}
$$

Using the Gamma reflection identity

$$
\Gamma(1+x)\,\Gamma(-x) = \frac{-\pi}{\sin(\pi x)}
\tag{5.24}
$$

it follows that

$$
I(q_\perp) = \pi \frac{\Gamma(1+\boldsymbol{\alpha})}{\Gamma(-\boldsymbol{\alpha})} \left(\frac{4\hbar^2}{q_\perp^2}\right)^{1+\boldsymbol{\alpha}} \,,
\tag{5.25}
$$

which matches (5.13), as desired (recall that $\boldsymbol{\alpha} \equiv -\mathrm{i}\alpha(s)$).

### 5.3 Saddle point, poles and classical bound states

With the analytic formula in hand, we can discuss some consequences of the result.

**Saddle.** First of all, recall that the eikonal approximation should break down for impact parameters of order of the Schwarzschild radius of the problem, where strong curvature effects are expected. In our case, $R_S = 2G_N\Lambda$, where $\Lambda$ is of order $\sqrt{\mathrm{Max}(s, m^2, M^2)}$ and represents the amount of mass available to form a black hole out of the rest masses and center of mass kinetic energy. In the ultra relativistic regime, $\Lambda \sim \sqrt{s}$. The two-dimensional Fourier transform $\int d^2 x_\perp$ above should thus be thought of as having a cut-off $|x_\perp| > R_S$, and the integral result should not be sensitive to $R_S$ scale physics.

Fortunately, it is a classic result [93, 94], which is immediate to verify. Firstly, for physical values of $s$, the integrand does not yield a divergence when $x_\perp \to 0$. Secondly, the integrand is dominated by a saddle, a critical impact parameter $x_\perp = b_*$ [93] given by

$$
|b_*| = \frac{\alpha(s)}{|q_\perp|} \,.
\tag{5.26}
$$

Using the explicit expression of $\alpha(s)$ given above, it is immediate to see that

$$|b_*| \simeq \frac{G_N \Lambda^2}{|q_\perp|} \gg G_N \sqrt{\Lambda} = R_S. \tag{5.27}$$

because we work in the limit of small momentum transfer $s/q_\perp^2 \gg 1$. Note that we did not have to assume a ultra-relativistic limit where $s \gg M^2, m^2$.

Plugged back the integrand, the saddle yields a pure phase and an inverse Jacobian, which can be calculated to be $1/J = \alpha(s)/b_* = \alpha(s)/t$. Therefore, on the saddle, the full amplitude behaves as

$$\mathcal{M}_{\mathrm{eik}}(q_\perp) \propto \sqrt{(p \cdot P)^2 - m^2 M^2} \, \frac{\alpha(s)}{t} \, \mathrm{e}^{\mathrm{i}\phi}, \tag{5.28}$$

which is precisely recognised to be, using (5.8), the four-point amplitude of (5.7), dressed with a phase. Anticipating on the following paragraph, note that this expression can not be used read the poles of the bound-state system. As we explain now, those originate from the region of integration $x_\perp \to 0$, to which the saddle is insensitive.

**Poles and zeroes.** Here, we review the analysis of Kabat and Ortiz [47], in order to set the stage for a similar analysis in the case of linearized Kerr in the next section. A remarkable feature of equation (5.25) is the presence of poles located at positive integers $\mathrm{i}\alpha(s) = n$ such that $n \in \mathbb{N}$, which are known to contain non-trivial information about the bound states of the system (see a related discussion in the review [95]). Generalising [47] to the unequal-mass case, we find that the poles are located at

$$s_n^{\mathrm{poles}} = m^2 + M^2 \pm \sqrt{4m^2M^2 - \frac{\hbar^2 n^2}{G^2} + \sqrt{\frac{\hbar^4 n^4}{G^4} + \frac{8m^2M^2\hbar^2 n^2}{G^2}}} \Big/ \sqrt{2}, \tag{5.29}$$

and correspond to the energies (squared) of the bound state system.

The location of these poles can also be read immediately off the integral expression (5.10), by noticing that the integral is regular unless $\alpha(s) = -\mathrm{i}n$. What cannot be read off from this expression are the location of the zeros of the integral; these are identified in the explicit expression in terms of Gamma functions. The zeros are located at positive integers $\mathrm{i}\alpha(s) = -n$ for $n > 0$ and are found to be given by

$$s_n^{\mathrm{zeros}} = m^2 + M^2 \pm \sqrt{4m^2M^2 - \frac{\hbar^2 n^2}{G^2} - \sqrt{\frac{\hbar^4 n^4}{G^4} + \frac{8m^2M^2\hbar^2 n^2}{G^4}}} \Big/ \sqrt{2}, \tag{5.30}$$

which again reduce to the zeros of [47] in the equal mass case.

The poles accumulate near the threshold points $s = (M \pm m)^2$, while the zeros are located in the complex plane at $s_n^{\mathrm{zeros}} = m^2 + M^2 + \mathrm{i}\Re(s_n^{\mathrm{zeros}})$. The bound states are those of the gravitational hydrogen atom, in a linearized $1/r$ potential.

## 6 Kerr background

Using the formalism developed in Section 4, we now compute the $1 \to 1$ scattering amplitude for a massive scalar in a linearized Kerr black hole space-time. The result is then

interpreted as the eikonal amplitude for $2 \to 2$ gravitational scattering of a massive scalar and a massive particle with infinite quantum spin. The tree-level/Born amplitude for this process was determined only recently [35–37], and we show that the $1 \to 1$ scalar amplitude on Kerr corresponds to the eikonal exponentiation of that result. Alternatively, one can view our amplitude on Kerr as providing an alternative derivation of the tree-level formula.

We also analyze the structure of the integrals in this spinning eikonal amplitude, finding a surprisingly rich KLT-like factorized form. The saddle points, poles, and zeros of the amplitude are also described.

## 6.1 The eikonal phase for Kerr

Although the $1 \to 1$ amplitude is a gauge invariant quantity, calculations are much simpler when the linearized background metric is expressed in harmonic coordinates. In such coordinates, the linearized Kerr metric admits a remarkable closed form which was first given by Vines [96]:

$$g_{\mu\nu}(x) = \eta_{\mu\nu} + \mathcal{P}_{\mu\nu\alpha\beta}\, h_a^{\alpha\beta}(x) \quad , \quad \mathcal{P}_{\mu\nu}{}^{\alpha\beta} = \delta_{(\mu}{}^{(\alpha}\delta_{\nu)}{}^{\beta)} - \frac{1}{2}\eta_{\mu\nu}\eta^{\alpha\beta} \, , \qquad (6.1)$$

$$h_a^{\alpha\beta}(x) := \left( u^\alpha u^\beta \cos(a\cdot\partial) + u^{(\alpha}\epsilon^{\beta)}{}_{\rho\lambda\mu}\, u^\rho\, a^\lambda\, \partial^\mu\, \frac{\sin(a\cdot\partial)}{a\cdot\partial} \right) \left( \frac{4G\,M}{r} \right) \, , \qquad (6.2)$$

where $u^\mu$ is the unit timelike vector for the source of Kerr, while $a^\mu$ is the (mass-rescaled) covariant spin vector,

$$a^\mu = \frac{1}{2M}\, \epsilon^\mu{}_{\nu\alpha\beta}\, u^\nu\, S^{\alpha\beta} \quad \Leftrightarrow \quad S^{\mu\nu} = M\, \epsilon^{\mu\nu}{}_{\alpha\beta}\, u^\alpha\, a^\beta \, , \qquad (6.3)$$

with $a\cdot u = a^\mu u_\mu = 0$, where $\epsilon_{\mu\nu\alpha\beta}$ is the (flat) 4d Levi-Civita symbol.

To evaluate the scattering amplitude for a scalar particle crossing this classical background, we first need to compute the Fourier transform of the projected metric tensor contributing to the eikonal phase (5.3). This is

$$\tilde{h}_a^{\alpha\beta}(q_\perp) = \left( u^\alpha u^\beta \cos(\mathrm{i}\, a\cdot q_\perp) - u^{(u}\epsilon^{\nu)}{}_{\rho\lambda\mu}\, u^\rho\, a^\lambda\, q_\perp^\mu\, \frac{\sin(\mathrm{i}a\cdot q_\perp)}{a\cdot q_\perp} \right) \left( \frac{16\pi G\,M}{q_\perp^2} \right) \, , \qquad (6.4)$$

where $q_\perp^\mu$ is orthogonal to $u^\mu$ and to the incoming momentum $p^\mu$ of the probe particle. Using equation (2.1) we can perform all scalar contractions on the support on the pole kinematics $q^2 = 0$ where[5] $\mathrm{i}\epsilon_{\mu\nu\rho\sigma}p^\mu P^\nu q^\rho a^\sigma = mM\gamma v\,(q\cdot a)$. As a result we obtain the following 4-point function underlying the eikonal phase for Kerr

$$A_4(q_\perp) = \frac{8\pi\, M^2\, m^2\, G\, \gamma^2(v)}{q_\perp^2} \left( (1+v)^2\, \mathrm{e}^{a_\perp\cdot q_\perp} + (1-v)^2\, \mathrm{e}^{-a_\perp\cdot q_\perp} \right) \, , \qquad (6.5)$$

where $P^\mu = Mu^\mu$. The quantity $A_4(q_\perp)$ represents the scattering amplitude for a test body $m$ gravitationally interacting at tree level with a massive object M with infinite quantum spin [38, 97]. Interestingly, due to its covariant form this also represents the scattering

---

[5]E.g. see eq.(44) in [36].

amplitude for spinning massive objects – and arbitrary mass ratios – first computed in this exponential form by Guevara, Ochirov and Vines [35] and subsequently confirmed with different methods[6] [36, 98]. As for the related eikonal phase, a straightforward computation gives

$$\chi_1(x_\perp) = -2\hbar \sum_\pm \alpha_\pm(s) \log(\mu \, |x_\perp \mp a_\perp|) \quad , \quad \alpha_\pm(s) := \frac{G \, m \, M \, (1 \pm v)^2 \, \gamma(v)}{2\hbar \, v} \, , \quad (6.6)$$

in agreement with the eikonal phase for Kerr computed in [36, 59].

Note that $\alpha_+(s) + \alpha_-(s) = \alpha(s)$. Explicit expressions in terms of $s$ can be unpacked from the definition of $\alpha(s)$ and the gamma factor in (5.11) and (5.12):

$$\alpha_\pm(s) = \pm \frac{G}{2\hbar} \left(s - m^2 - M^2\right) + \frac{\alpha(s)}{2} \tag{6.7}$$

Using that $\alpha(s)$ is explicitly invariant under crossing symmetry $s \leftrightarrow u = 2m^2 + 2M^2 - s$ (at small $t$), we also discover that $\alpha_+(s)$ and $\alpha_-(s)$ switch under crossing:

$$\alpha_+(s) = \alpha_-(2m^2 + 2M^2 - s) \tag{6.8}$$

This relation is going to be useful later when we check the crossing symmetry of the Kerr eikonal amplitude, to which we turn now.

## 6.2 The eikonal amplitude on a Kerr background

At this point, we apply our prescription (4.20) to the $1 \to 1$ amplitude on Kerr to extract a candidate $2 \to 2$ eikonal scattering amplitude. Following recent interpretations of Kerr in the context of scattering amplitudes [35–38, 56–64], the natural candidate eikonal amplitude is for scattering between a massive scalar and a massive particle with infinite quantum spin. This takes the form:

$$\mathcal{M}_{\text{eik}}(q_\perp) = -\frac{4\mathrm{i} \, \sqrt{(p \cdot P)^2 - m^2 M^2}}{\hbar^2} \int \mathrm{d}^2 x_\perp \, \mathrm{e}^{-\mathrm{i} q_\perp \cdot x_\perp / \hbar} \left( |x_\perp - a_\perp|^{2\boldsymbol{\alpha}} \, |x_\perp + a_\perp|^{2\boldsymbol{\beta}} - 1 \right) \, , \tag{6.9}$$

where we have omitted the mass regulator and have introduced the shorthand notation

$$\boldsymbol{\alpha} := -\mathrm{i}\,\alpha_+(s), \quad \boldsymbol{\beta} := -\mathrm{i}\,\alpha_-(s) \, . \tag{6.10}$$

This integral therefore generalises the classic eikonal amplitude for scalars (5.9) to a case that captures some spin effects. The calculation of the integral is much more involved than the original spinless case and we devote the rest of this section to its determination by means of a KLT-like factorization, analogous to the one presented above. We performed several checks on the final result (presented hereafter); we checked its explicit invariance under crossing symmetry (this is a very non-trivial check), the location of its poles, and its short $q$ behaviour. Together, these provide very strong evidence that our result is correct.

---

[6]See also [63] for a recent interesting account on this topic.

Now, for the non-trivial part of the amplitude we want to compute an integral of the form

$$A(q_\perp, a_\perp) = \int \mathrm{d}^2 z \, \mathrm{e}^{-\mathrm{i}\,(qz + \bar{q}\bar{z})/2\hbar} \, |z - a_\perp|^{2\alpha} \, |z + a_\perp|^{2\beta} \ . \tag{6.11}$$

After shifting $z \to z - a_\perp$ and rescaling $z \to (2a_\perp)z$, the calculation is reduced to the following integral

$$I(q_\perp) = \int \mathrm{d}^2 z \, \mathrm{e}^{-\mathrm{i}\,(qz + \bar{q}\bar{z})/\hbar} \, |z - 1|^{2\alpha} \, |z|^{2\beta} \ , \tag{6.12}$$

which is related to $A$ by $A(q_\perp, a_\perp) = |2a_\perp|^{2 + 2\alpha + 2\beta} \, \mathrm{e}^{-\mathrm{i}(qa + \bar{q}\bar{a})/2} \, I(2q_\perp a_\perp))$.

Applying a similar method to the one used in section 5.2 above, we found that the integral can be decomposed as sums of bilinears of the following three elementary integrals, whose explicit expression in terms of confluent hypergeometric functions is provided in appendix A:

$$I_1 = \int_0^1 \mathrm{d}w \, \mathrm{e}^{-\mathrm{i}\,qw/\hbar} \, w^\alpha \, (1 - w)^\beta \ , \tag{6.13}$$

$$I_2 = \int_1^\infty \mathrm{d}w \, \mathrm{e}^{-\mathrm{i}\,qw/\hbar} \, w^\alpha \, (w - 1)^\beta \ , \tag{6.14}$$

$$I_3 = \int_{-\infty}^0 \mathrm{d}w \, \mathrm{e}^{-\mathrm{i}\,qw/\hbar} \, (-w)^\alpha \, (1 - w)^\beta \ . \tag{6.15}$$

These three functions, $I_1, I_2$ and $I_3$ are not all independent. One can integrate $\mathrm{e}^{-\mathrm{i}qz} z^a (1 - z)^b$ along the real axis, which gives a vanishing answer, since the contour can be closed with a (vanishing) arc at infinity (upper or lower half plane, depending on the sign of $\Im q$). If $\Im q < 0$, we have the following relation:[7]

$$\mathrm{e}^{\mathrm{i}\pi\alpha} I_3 + I_1 + \mathrm{e}^{-\mathrm{i}\pi\beta} I_2 = 0 \ . \tag{6.16}$$

It is an instance of Kummer's relations; see (A.2).

**The final result of this section is that**

$$I(q_\perp) = -(\sin(\pi\alpha)\tilde{I}_1 I_3 + \sin(\pi\beta)\tilde{I}_2 I_1 + \sin(\pi(\alpha + \beta))\tilde{I}_2 I_3) \tag{6.17}$$

where the tilde above the $I_j$ integrals means that they are evaluated at complex conjugate argument $\bar{q}$:

$$\tilde{I}_j := I_j(\bar{q}_\perp) \, , \tag{6.18}$$

(the other arguments, $\alpha, \beta$ are not conjugated). This formula can also be recast in the following form, using Kummer's relation (6.16) above:

$$I(q_\perp) = -(\tilde{I}_1, \tilde{I}_3) \cdot S \cdot (I_1, I_3)^{\mathrm{T}} \, , \tag{6.19}$$

---

[7]In string theory, this property helps to decrease the number of basis amplitudes from $(n - 2)!$ to $(n - 3)!$ [99–101], and is related to the famous Bern-Carrasco-Johansson relations [102]. Here, the number of points $n$ is fixed by the geometry of space-time (Kerr, Schwarzschild, shockwave) of the integrand and does not relate to the number of particles involved in the scattering. It would be however interesting to study this point further.

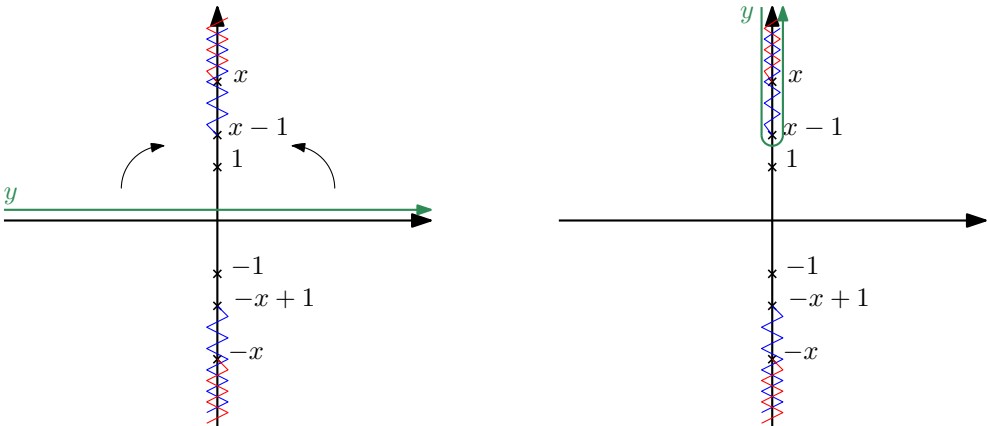

**Figure 5**. Contour deformation. The real contour (green) for $y$ is folded up along the branch cut. On each side of the branch, the integral picks up a phase, as for the calculation in Schwarzschild background.

where $S$ is a momentum-kernel-like [103] matrix defined by

$$S = -\frac{1}{2\mathrm{i}} \begin{pmatrix} 1 - e^{2\mathrm{i}\pi\boldsymbol{\beta}} & -e^{\mathrm{i}\pi\boldsymbol{\alpha}}\left(-1 + e^{2\mathrm{i}\pi\boldsymbol{\beta}}\right) \\ -e^{\mathrm{i}\pi\boldsymbol{\alpha}}\left(-1 + e^{2\mathrm{i}\pi\boldsymbol{\beta}}\right) & 1 - e^{2\mathrm{i}\pi(\boldsymbol{\alpha}+\boldsymbol{\beta})} \end{pmatrix} . \tag{6.20}$$

We now turn to the details of the calculation.

**Details of the calculation.** Our starting point (6.12) can be rewritten as:

$$I(q_\perp) = \int \mathrm{d}x \, \mathrm{d}y \, e^{-\mathrm{i}\,(qz + \bar{q}\bar{z})/\hbar} \, (z - 1)^{\boldsymbol{\alpha}} \, (z)^{\boldsymbol{\beta}} \, (\bar{z} - 1)^{\boldsymbol{\alpha}} \, (\bar{z})^{\boldsymbol{\beta}} . \tag{6.21}$$

The basics of the calculation are the same as in sec. 5.2. Assuming $\Im q < 0$, the $y$-contour needs to be closed in the upper-half plane, to ensure vanishing of arcs at infinity. There are branch cuts starting at $y = \pm \mathrm{i}x$ and $y = \pm \mathrm{i}(x - 1)$. The contour $y \in \mathbb{R}$ is folded along a contour on the vertical axis $y = \mathrm{i}\tilde{y}$ with $\tilde{y}$ real. The difference between left- and right- contours, with the sign for opposite orientations gives factors of $\sin(\pi\boldsymbol{\alpha})$, $\sin(\pi\boldsymbol{\beta})$ or $\sin(\pi(\boldsymbol{\alpha}+\boldsymbol{\beta}))$ depending on the ordering of the branch point singularities $\pm x$ and $\pm(x-1)$. An example of the contour folding is depicted in fig. 5, for which $x > 1$.

As we shall see below, there are essentially two different cases, which correspond to whether the first cut is associated to the phase $e^{\pm \mathrm{i}\pi\boldsymbol{\alpha}}$ or $e^{\pm \mathrm{i}\pi\boldsymbol{\beta}}$.

For definiteness, let us start with $x > 1$, which corresponds to the branch cut arrangement depicted in figure 5, and 6, a). Along the folded contour on the imaginary axis, the $\mathrm{d}\tilde{y}$ integral gives two contributions, with phase factors $\sin(\pi\boldsymbol{\beta})$ for $\tilde{y} \in [x - 1; x]$ along the blue cut, and $\sin(\pi(\boldsymbol{\alpha} + \boldsymbol{\beta}))$ along the red and blue cuts (we shall state this in equations below) for $\tilde{y} > x$.

When $x$ decreases towards $x = 1/2$, the upwards and downwards blue cuts cross each other, see fig. 6, b). The phases annihilate in the interval where the cuts are on top of each other, $\tilde{y} \in [x, 1 - x]$, and the net result is that a blue cut now extends from $1 - x > 0$ to

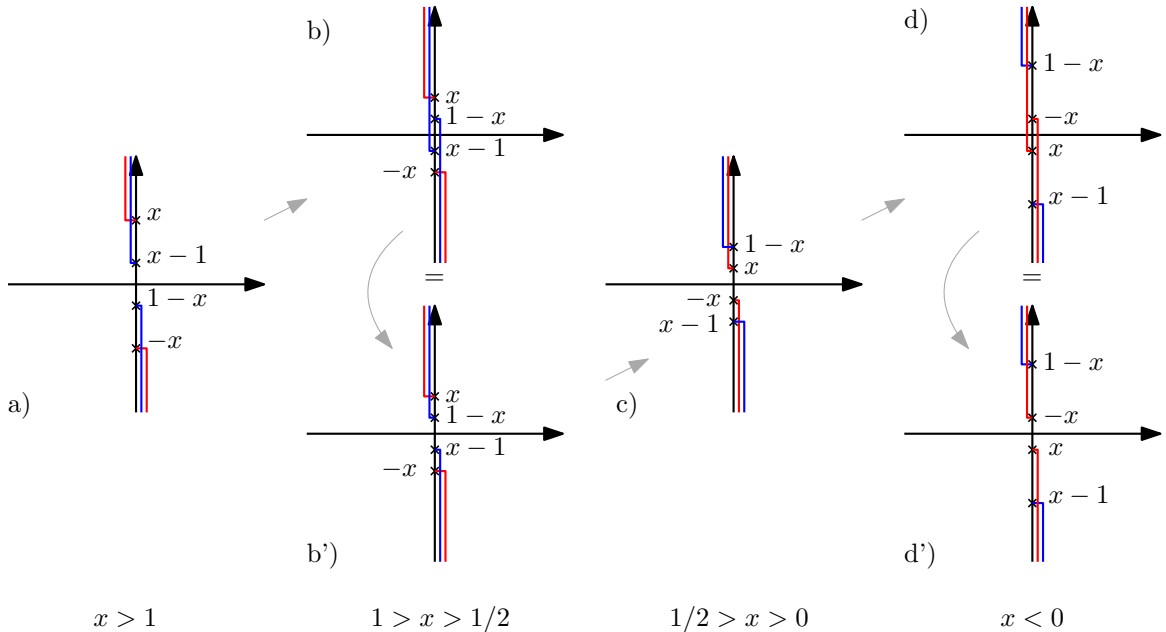

**Figure 6**. Change of branch cuts. We have adopted a different graphical depiction of the cuts for the sake of clarity, but their position relative to the vertical axis is irrelevant here. Only the fact that the contour passes on the left and right of those cuts matters.

positive infinity, as in fig. 6, b'). Therefore, this does not change the structure of the phase factors which remain $\sin(\pi\boldsymbol{\beta})$, $\sin(\pi(\boldsymbol{\alpha}+\boldsymbol{\beta}))$. This is similar to what is described in fig. 3.

The blue and red cut exchange position when $x$ crosses $1/2$. Thus, for $x < 1/2$, we have a different phase factor, now given by $\sin(\pi\boldsymbol{\alpha})$ for $\tilde{y} \in [x; 1-x]$ and $\sin(\pi(\boldsymbol{\alpha}+\boldsymbol{\beta}))$ for $\tilde{y} > 1 - x$. When $x$ eventually becomes negative, the two branch points $\pm \mathrm{i}x$ exchange location but this does not change the phase factors, which remain $\sin(\pi\boldsymbol{\alpha})$, $\sin(\pi(\boldsymbol{\alpha}+\boldsymbol{\beta}))$, just like above and in fig. 3; see fig. 6, d) and d').

The conclusion of this discussion is that the integration domain in $x$ needs to be divided in two regions : $x \geq 1/2$ (graphs a), b') ) and $x \leq 1/2$ (graphs c), d')), and the full integral is given by a sum of two terms

$$I(q_\perp) = I_+ + I_- \ , \tag{6.22}$$

defined below.

**First case: $x \geq 1/2$.** The discussion above helped to understand the structure of the integral we started from, restricted to $x > 1/2$, which is given by:

$$I_+ = -2 \int_{1/2}^{\infty} \left( \sin(\pi\boldsymbol{\beta}) \int_{|x-1|}^{x} f(u)\,g(v)\,\mathrm{d}y + \sin(\pi(\boldsymbol{\alpha}+\boldsymbol{\beta})) \int_{x}^{\infty} f(u)\,g(v)\,\mathrm{d}y \right) \mathrm{d}x \ , \tag{6.23}$$

where $f(u)$ and $g(v)$ build up the single valued integrand (i.e. the integrand stripped out of the phases associated to $(-1)^{\boldsymbol{\alpha}}, (-1)^{\boldsymbol{\beta}}$) factors), defined by

$$f(u) = \mathrm{e}^{-\mathrm{i}\,uq/\hbar}\,|u-1|^{\boldsymbol{\alpha}}\,|u|^{\boldsymbol{\beta}} \ , \tag{6.24}$$

$$g(v) = \mathrm{e}^{-\mathrm{i}\,v\bar{q}/\hbar}\,|v-1|^{\boldsymbol{\alpha}}\,|v|^{\boldsymbol{\beta}} \ . \tag{6.25}$$

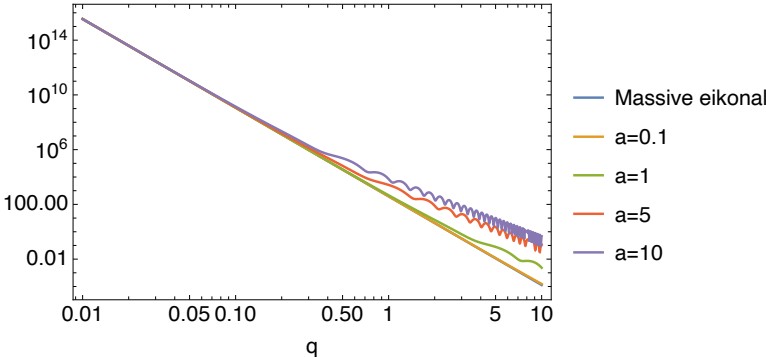

**Figure 7**. Blue: absolute value of the scalar eikonal amplitude as defined in (5.13). Colors: absolute value of Kerr eikonal amplitude $A(q_\perp, a_\perp) = |2a|^{2+2\boldsymbol{\alpha}+2\boldsymbol{\beta}} I(q_\perp a_\perp)$, obtained from (6.17) with varying $a_\perp$ as indicated by $a = ...$ on the figure. For $a \to 0$, the Kerr eikonal amplitude descends to the Schwarzschild eikonal amplitude. At larger $q$, the Kerr eikonal amplitude is the sum of two decaying oscillatory terms which explains that the absolute values displays oscillations. The absolute values of the scalar eikonal amplitude kills these oscillations. This plot is obtained for complexified kinematics $\alpha, \boldsymbol{\beta}$ real constants of order 1.

Again, we slightly abuse notation here, by writing the integrand in terms of $u, v = x \pm \tilde{y}$ and the measure and boundary in terms of $x, \tilde{y}$. The reader annoyed by this should consider that $u$ and $v$ are functions of $x, \tilde{y}$, $u := u(x, \tilde{y}) = x - \tilde{y}$ and $v := v(x, \tilde{y}) = x + \tilde{y}$. The factor of $-2$ comes from the 2i from the sine function and an i for the measure $\mathrm{d}y = \mathrm{i}\mathrm{d}\tilde{y}$.

**Second case: $x \leq 1/2$.** The other relevant domain of the $x$-integration yields

$$I_- = -2 \int_{-\infty}^{1/2} \left( \sin(\pi\boldsymbol{\alpha}) \int_{|x|}^{1-x} f(u)\,g(v)\,\mathrm{d}\tilde{y} + \sin(\pi(\boldsymbol{\alpha}+\boldsymbol{\beta})) \int_{1-x}^{\infty} f(u)\,g(v)\,\mathrm{d}\tilde{y} \right) \mathrm{d}x \ . \tag{6.26}$$

The last stage is to show that the integrals above can indeed be written as separate integrals of $u$ and $v$. We will show how this happens after collecting the pieces of integration domain corresponding to the same phase factors.

Consider first the case of $\sin(\pi\boldsymbol{\beta})$, which is found only in $I_+$. The corresponding domain of integration is $x \geq 1/2, x \geq \tilde{y} \geq |x - 1|$. Carefully drawing this domain, using a picture similar to fig. 4, yields that this domain is just $v \geq 1, 1 \geq u \geq 0$. Likewise, the contribution of the $\sin(\pi\boldsymbol{\alpha})$ term in $I_-$ can be seen to be given by an integral over the following domain, $0 \leq v \leq 1, u \leq 0$. Finally, the term $\sin(\pi(\boldsymbol{\alpha}+\boldsymbol{\beta})$ receives contributions from both $I_+$ and $I_-$, which, once pieced together, corresponds to the domain $v > 1, u < 0$. Rewriting (6.22) using this analysis, adding a factor of $1/2$ for the Jacobian $\mathrm{d}x\,\mathrm{d}\tilde{y} = 1/2\mathrm{d}u\,\mathrm{d}v$ and the definitions of $I_1, I_2, I_3$ above yields (6.17).

**Direct checks.** We performed two direct checks on this formula: we checked its invariance under crossing and its small $a_\perp$ limit. In the $a \to 0$ limit, fig.7 shows that the Kerr eikonal amplitude descends to the Schwarzschild eikonal amplitude, which is trivially expected by comparing eq.(6.12) to eq.(5.10).

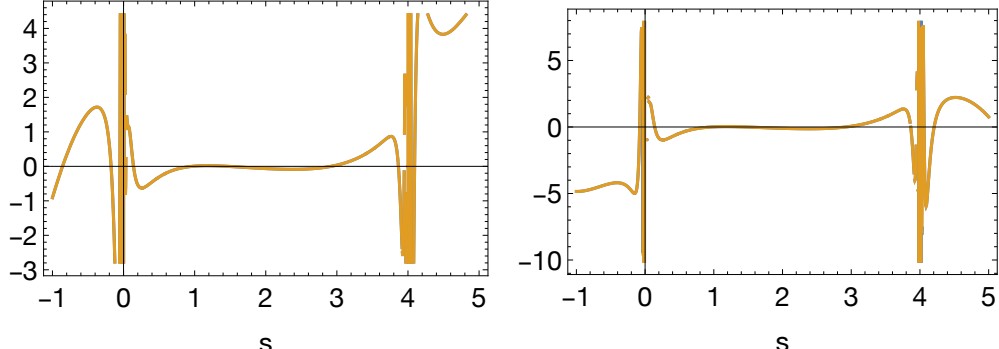

**Figure 8**. Check of crossing. The plots show the real (left) and imaginary (right) parts of essentially the function $I$ which we evaluated in (6.17). For each plots, *two curves are plotted which exactly overlap*, $I(q,s)$ and $e^{2i\Re(q)}I(-q, 4-s)$ : the curves are indistinguishable and this shows that the result of the lengthy KLT-like calculation exhibits crossing symmetry of $I$ shown in eq.(6.27).

The most non-trivial check is that of crossing symmetry. Under $s \leftrightarrow u = 4m^2 - s$, $\alpha_+$ and $\alpha_-$ get exchanged, as was explained in (6.8). This shows immediately that the eikonal Kerr amplitude (6.11) is crossing symmetric under $s \leftrightarrow u = 4m^2 - s$, up to a change of sign of $q_\perp$. At the level of the function $I(q_\perp)$ defined in (6.12), we have a similar transformation, which is given by

$$I(q_\perp, s) = e^{-2i\Re(q_\perp)}I(-q_\perp, 4-s) \tag{6.27}$$

where we have introduced the explicit dependence on $s$ of the integral $I(q_\perp)$ for obvious notational purposes. However, the final expression (6.17) is not obviously symmetrical under $\boldsymbol{\alpha} \leftrightarrow \boldsymbol{\beta}$, therefore, checking this symmetry is highly non-trivial. We have verified this explicitly in mathematica, and we present in fig. 8, some plots representing the crossing symmetry of our final answer (6.17). As a matter of fact, this property delicately relies on the precisely chosen phases, and does not hold for other combinations.

These checks gives us strong confidence that our result is indeed correct.

## 6.3 Saddle point, poles and classical bound states

A few things can be said about the integral (6.17). Firstly, one can read off the poles of the amplitude without having to compute the integral. Secondly, a saddle-point analysis gives an expansion at small spin of the amplitude.

**Saddle-point.** The saddle-point is obtained as before by solving the equations

$$\frac{\partial}{\partial z}\left((qz + \bar{q}\bar{z}) + \alpha_- \log(|z+a|^2)) + \alpha_+ \log(|z-a|^2)\right) = 0 \tag{6.28}$$

This gives a second-order equation in $z$, with two solutions. One of these two solutions reduces to the ACV saddle (5.26) when $a \to 0$ while the other one runs away to $x_\perp \to 0$. This region of the integral should not carry information about the physical scattering regime (cf., discussion of scales in the beginning of sec. 5.3), so we will not investigate it here, this will be sufficient to reproduce the small $a$ limit, i.e., the Schwarzschild eikonal.

Because neither the details of the calculation nor the explicit expressions – which at intermediate and final stages are unwieldy composed expressions of long square roots coming from solving (6.28) – we shall simply provide a few key results.

We can, however, Taylor expand the result in the dimensionless variable $aq$, which we re-express in a real-valued notation, such that $\bar{q}a = q \cdot a + \mathrm{i}q \wedge a$. The result of a lengthy calculation easily yields the first few terms of this expansion, which we provide below for illustrative purposes

$$A(s,t) = \left(\frac{\alpha_-}{q^2}\right)^{1-2\mathrm{i}\alpha_-}\left(1 + (a \cdot q)\frac{2\mathrm{i}\alpha_+}{\alpha_-} + (a \cdot q)^2\frac{\alpha_+{}^2(1 - 2\alpha_-{}^2 + \mathrm{i}\alpha_-) - \alpha_-{}^2 - \mathrm{i}\alpha_-{}^3)}{\alpha_-{}^4}\right.$$
$$\left. + (a \wedge q)^2\frac{(1 + \mathrm{i}\alpha_-)(\alpha_-{}^2 - \alpha_+{}^2)}{\alpha_-{}^4} + O(a^3)\right) \quad (6.29)$$

We have not characterised the nature of this expansion in $aq$ but it is tempting to hypothesize that it captures some sort of gravito-electric and gravito-magnetic [104] remnants of the interactions in the linearized Kerr background.

**Poles.** The non-analyticities of the integral (6.11) are easy to read off the integral itself. Just like in the case of the shockwave calculation above, $\mathcal{M}_{\mathrm{eik}}(q_\perp)$ can blow up if and only if the argument of the functions $|x_\perp \pm a_\perp|$ goes to zero and at the same time the exponent goes to a negative value. If we choose $+a$ first for definiteness, in this region, the integral simplifies to

$$\mathcal{M}_{\mathrm{eik}}(q_\perp)\Big|_{x_\perp \to a_\perp} \simeq \mathrm{e}^{-\mathrm{i}\,q_\perp \cdot a_\perp}\,|2a_\perp|^{-2\mathrm{i}\alpha_+(s)}\int \mathrm{d}^2x_\perp\,\mathrm{e}^{-\mathrm{i}\,q_\perp \cdot x_\perp}\,|x_\perp|^{-2\mathrm{i}\alpha_-(s)} \quad (6.30)$$

$$\simeq \mathrm{e}^{-\mathrm{i}q_\perp \cdot a_\perp}\,|2a_\perp|^{-2\mathrm{i}\alpha_+(s)}\,\frac{\Gamma(1 + \mathrm{i}\alpha_-(s))}{\Gamma(-\mathrm{i}\alpha_-(s))}\,\frac{\mathrm{e}^{\mathrm{i}\phi}}{q_\perp^2}$$

up to small corrections, which indeed provides poles at negative values of $\alpha_-(s)$. Proving that the integral is regular in $\alpha_-, \alpha_+$ away from those regions goes as follows. Firstly, for large values of $x_\perp$, the integrand reduces to the original 't Hooft integrand up to small corrections. But we know that the non-analyticities in the 't Hooft integral come from the region $x_\perp \to 0$, hence the portion of integration corresponding to large $x_\perp$ yields analytic contributions of $\alpha_-, \alpha_+$. Furthermore it is immediately clear that the finite domain between large $x_\perp$ and $x_\perp \to \pm a_\perp$ does not bring any non-analyticities (we consider the integral of a continuous function over a compact domain), which achieves to prove that the only poles are those of (6.30) and the ones with $\alpha_-$ and $\alpha_+$ interchanged.

The poles are then defined by $\mathrm{i}\alpha_+(s) = n$ and $\mathrm{i}\alpha_-(s) = n$ with $n$ a positive integer. What were zeros in the shockwave case, that is negative values of $n$, are less straightforward to interpret now, because the whole integral is not given anymore by the expressions (6.30), which are just local expressions. It might be that the rest of the integration domain makes the integral non-zero overall. Therefore, we shall refer to these points as *improper zeros*, corresponding $\mathrm{i}\alpha_+(s) = -n$ and $\mathrm{i}\alpha_-(s) = -n$ for $n > 0$.

Firstly, let us emphasize that the location of the new poles does not depend on $a$. While it is clear that, when $a = 0$, the linear Kerr eikonal amplitude reduces to the 't Hooft

eikonal amplitude, this does not happen by having poles which are smoothly connected to one another. The change is more violent, and the *residues* of the Kerr poles vanish at $a = 0$, while the 't Hooft pole become actual poles only when $a = 0$, not before. Yet, our amplitude knows about the $+$ and $-$ polarisations which correspond to the $+$ and $-$ terms in the Kerr amplitude.[8] It would be interesting to investigate this further.

Now, let us describe briefly the location of the poles and improper zeros of the amplitude, depicted in figure 9. Because $\alpha_+$ and $\alpha_-$ exchange their locations under $s \leftrightarrow 4m^2 - s$, we will only describe the case of the poles and improper zeros of Firstly, the poles of $i\alpha_\pm(s) = n$ are related by crossing $s \leftrightarrow 4m^2 - s$. They accumulate near $s = 0, 4m^2$, but, contrary to the case of the scalar massive eikonal, they have a small imaginary part (which decreases with $1/n$). Secondly, the improper zeros of $i\alpha_\pm(s) = -n$ split in two series : one series are complex conjugate to the zeros and accumulate near $s = 0, 4m^2$, and the other series is located on the imaginary axis $\Re(s) = 2m^2$, close to the standard scalar massive eikonal zeros. Since they are related by $s \leftrightarrow 4m^2 - s$, the series on the positive imaginary axis corresponding to $i\alpha_+(s) = -n$ is just the mirror of $i\alpha_-(s) = -n$ on the negative imaginary axis.

Therefore we observe that the massive scalar eikonal poles on the real axis near $s = 0, 4m^2$ are 'lifted' to pairs of complex conjugates pole/improper zeros, while the 't Hooft zeros at $\Re(s) = 2m^2$ are also slightly lifted with just improper zeros. Since the 'improper zeros' are not zeros of the amplitude, this poses no problem per se, but it would be interesting to understand exactly the analytic structure here. Additionally, all of the poles are complex, suggesting that the corresponding bound-states are unstable, a feature reminiscent of superradiance instabilities[9] for scalar perturbations around a Kerr space-times. A complete solution to the gravitational bound-state problem on Kerr has been recently investigated in [105, 106]. Presumably our bound states are a sub-sector of their full solution, and would be interesting to investigate this issue further in the future.

## 7 Discussion

In this paper, we proposed a covariant alternative to computing the $2 \to 2$ gravitational eikonal scattering amplitude in QFT in terms of a $1 \to 1$ scattering amplitude in curved space-time far from the source. This generalizes the much earlier observation of 't Hooft, linking ultra-relativistic eikonal scattering of scalars to $1 \to 1$ scattering on a shockwave space-time. We tested our proposal in Schwarzschild and Kerr space-times, finding complete agreement with eikonal scattering of massive scalars in the first case, and an amplitude which exponentiates the Born amplitude for scattering of a massive scalar and a massive particle with infinite spin in the second case.

There are several interesting open questions and future directions raised by this work; we discuss some of them here.

---

[8] We thank Alexander Ochirov for a discussion on this point.

[9] We are grateful to Riccardo Gonzo for enlightening conversations on this topic.

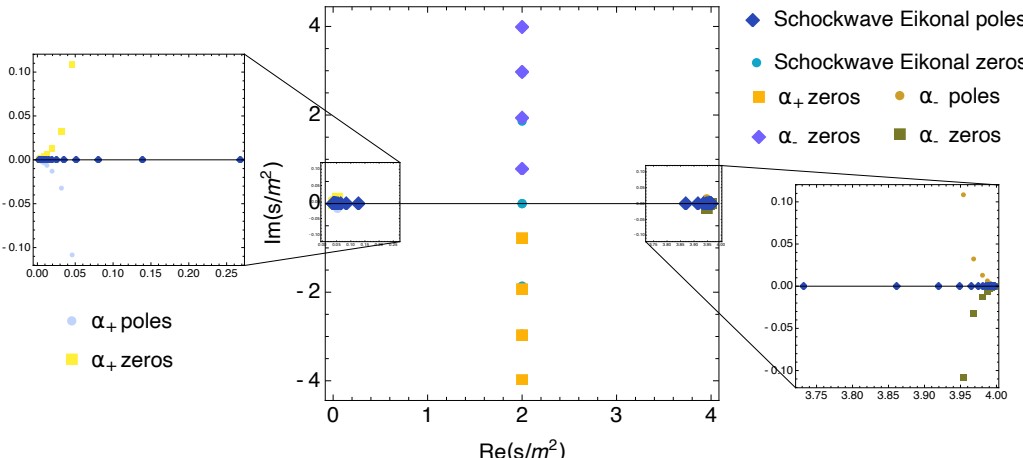

**Figure 9**. Near the extremities, $s = 4m^2$ and $s = 0$, the scalar massive eikonal poles split up into pairs of complex conjugate poles/improper zeros, near $s = 0$ for $\alpha_-$ and near $s = 4m^2$ for $\alpha_+$. On the axis $\Re(s) = 2m^2$ we find improper zeros for $\alpha_+$ on the positive imaginary axis, and $\alpha_-$ on the negative imaginary axis, near the locations of the Schwarzschild eikonal zeros which span the whole positive and negative imaginary axis.

**Proving the proposal.**   We showed that the $1 \to 1$ scattering amplitude in any linearized stationary space-time has the structural form of a $2 \to 2$ eikonal scattering amplitude. While our results for Schwarzschild and Kerr support the proposal that this really is *the* eikonal amplitude for a $2 \to 2$ scattering process with small momentum transfer, we have certainly not proved that this proposal is generally true. For instance, our computation in Kerr produced the exponentiation of the amplitude found in [36, 37], but the exponentiation of this amplitude has not yet been computed order-by-order in field theory, apart from the leading order term in the eikonal exponentiation [40].

To truly prove our proposal in full generality, one would require a first-principles derivation of the non-perturbative background space-time from an infinite resummation of ladder diagrams. At linear level, it is well known that space-times like Schwarzschild can be recovered by summing Feynman diagrams between a probe and source (cf., [107–110]). It would be interesting to relate these resummations to the eikonal resummation.

**Spinning vs unspinning probe.**   In our calculation in the Kerr space-time, only the background is spinning. In [36], the tree-level amplitude $A_4$ was obtained for the $2 \to 2$ scattering amplitude where *both* particles have spins $a_1, a_2$. However, because Kerr is really the motion of a minimal coupled infinite spin particle [36, 37], in the process the spins exponentiate and sum up in such a way that the end result depends only on the total spin $a = a_1 + a_2$ at leading order in the gravitational coupling.[10]  Similar calculations, including a spinning probe particle have been studied very recently[11] [111, 112].

For another spinning object with finite-size effects not captured by the Kerr metric, such as a neutron star, spin effects do not exponentiate. One could imagine computing

---

[10]We thank Alexander Ochirov for a discussion on this point.
[11]AC thanks Justin Vines for sharing a preliminary version of [111].

a $1 \rightarrow 1$ scattering amplitude in the space-time of a neutron star; our proposal suggests that – at least in a stationary approximation of the star – this amplitude will exhibit eikonal exponentiation, but the spin effects themselves may not exponentiate. It would be interesting to attempt such a scattering amplitude calculation for a neutron star, or indeed any space-time with finite size effects.

**Higher-point amplitudes.** In this paper, our focus has been on $1 \rightarrow 1$ amplitudes in curved space-time and their relation to $2 \rightarrow 2$ scattering in the leading eikonal limit. However, one could also compute higher-point amplitudes using the general framework of QFT in curved space-times. This will obviously introduce further complications: we were able to avoid strong-field effects (e.g., particle creation) that would have spoiled the existence of a S-matrix for $1 \rightarrow 1$ scattering by localizing the boundary term for the amplitude far from the source. Higher-point amplitudes will not be pure boundary terms, so finding a consistent way to avoid strong field effects will be more subtle.

Nevertheless, there are clear reasons to consider such higher-point amplitudes. For instance, it is natural to propose that $1 \rightarrow 2$ scattering in a curved space-time with an emitted graviton will correspond to small-angle $2 \rightarrow 3$ scattering with the emission of gravitational radiation. This is precisely the context of gravitational wave emission from binary collisions.

This idea of capturing 'eikonal with emission' from scattering on curved backgrounds has already been studied in the context of ultra-relativistic scattering by using shockwave backgrounds in gravity [85, 113] and QED [51]. Here, there are no ambiguities since the shockwave background admits an S-matrix, so there remains work to be done to extend these ideas to generic stationary backgrounds.

**Relation to string theory amplitudes and twisted intersection theory.** It would not have escaped the eye of the reader accustomed to string theory amplitudes that both the massive scalar eikonal and the Kerr eikonal amplitude are reminiscent of string theory amplitudes. An argument due to Verlinde and Verlinde [46] to explain the string-like structure of the old 't Hooft result is that the quantum gravity path integral, in the eikonal limit, should reduce to a topological-2d sigma model in the transverse plane. String like amplitudes are then obtained in the specific shockwave background by determining the phase shifts of the amplitude. It would be interesting to reproduce this calculation in our case, and we leave this to future work. Overall, we now have a possible qualitative explanation for the resemblance of this amplitude to string-like amplitudes, which should therefore expect to hold generically.[12] This story however is a little not fully satisfactory because the eikonalisation in the small $t$ regime is a phenomenon more generic than gravity. One might be tempted to speculate that the existence of KLT relations (to which we come to shortly after) might suggest that those models have intrinsically something to do with closed strings, hence gravity, but QED is also know to eikonalize and has the exact same structure in terms of products of Gamma functions and 2d integrals, see e.g. [114, (9)].

---

[12]Note that in higher dimensions, the transverse plane would be $D - 2 \geq 3$ dimensional and we loose the immediate string-like form. Since membrane amplitudes are not a well defined concept, we cannot speculate further.

However, it is still interesting to wonder about the physical nature of the single copy of the eikonal amplitudes, which would be a 1d effect of some sort. It would be interesting to understand these points further.

Another aspect of this study is the existence of a twisted (co)homology behind those integrals (see [115]). Precisely because the shockwave and the Kerr eikonal amplitude assume this string-like form, a formalisation of the KLT calculations can be immediately done. In this context, the three integrals, two of which are independent are a basis of amplitudes and the momentum-kernel-like matrix $S$, or rather its inverse [116], represents the intersection matrix between the twisted cycles. Likewise, the Kummer relations (6.16), (A.2) are nothing but the vanishing of a boundary twisted cycle. Contrary to the usual case of string theory, where we integrate rational functions against the multivalued form $\omega = x^s(1-x)^t dx$ and poses no problem of convergence, here we look at a Fourier transform and the exponential allows convergence only in one half-plane. Therefore, only one vanishing relation can be written, and not two.

## Acknowledgments

We are grateful to Nava Gaddam, Riccardo Gonzo, Alexander Ochirov, Donal O'Connell, Pierre Vanhove and Justin Vines for helpful conversations. We also thank Alexander Ochirov for comments on a draft. TA is supported by a Royal Society University Research Fellowship and by the Leverhulme Trust (RPG-2020-386). AC is supported by the Leverhulme Trust (RPG-2020-386).

## A Confluent hypergeometric functions

The integrals $I_1$, $I_2$ and $I_3$ defined in (6.13) can be expressed in terms of the $M$ and $U$ confluent hypergeometric functions, which assume the following integral representations (see for instance [117, chap. 13])

$$M(a,b,z) = \frac{\Gamma(b)}{\Gamma(a)\,\Gamma(b-a)} \int_0^1 e^{zt}\, t^{a-1}\,(1-t)^{b-a-1}\, dt\,, \quad \Re b > \Re a > 0$$

$$U(a,b,z) = \frac{1}{\Gamma(a)} \int_0^\infty e^{-zt}\, t^{a-1}\,(1+t)^{b-a-1}\, dt\,, \quad \Re a > 0$$

(A.1)

They obey the following relation, known as one of Kummer's relation:

$$U(a,b,z) = \frac{\Gamma(1-b)}{\Gamma(a-b+1)} M(a,b,z) + \frac{\Gamma(b-1)}{\Gamma(a)} z^{1-b} M(a-b+1, 2-b, z)\,.$$

(A.2)

We then immediately obtain

$$I_1 = \int_0^1 dw\, e^{-iqw}\, w^\alpha\,(1-w)^\beta = \frac{\Gamma(1+\alpha)\Gamma(1+\beta)}{\Gamma(\beta+\alpha+2)} M(1+\alpha, 2+\alpha+\beta, -iq) \quad \text{(A.3)}$$

$$I_3 = \int_{-\infty}^0 dw\, e^{-iqw}\, (-w)^\alpha\,(1-w)^\beta = \Gamma(\alpha+1)\, U(\alpha+1, \alpha+\beta+2, -iq) \quad \text{(A.4)}$$

Using the monodromy relation (6.16), which is nothing but the Kummer relation written above in (A.2), we finally get

$$I_2 = \int_1^\infty \mathrm{d}w\, e^{-iqw}\, w^\alpha\, (w-1)^\beta = \frac{\pi}{\sin(\pi(\alpha+\beta))} \left( \frac{\Gamma(1+\beta)}{\Gamma(\alpha+\beta+2)\Gamma(-\alpha)} M(1+\alpha, 2+\alpha+\beta, -iq) - \right.$$
$$\left. \frac{(iq)^{-\alpha-\beta-1}}{\Gamma(-\alpha-\beta)} M(-\beta, -\alpha-\beta, -iq) \right) \quad (A.5)$$

Note that Mathematica expresses $I_1$ in terms of `Hypergeometric1F1Regularized[a,b,z]` function, which is a confluent hypergeometric function denoted $\mathbf{M}(a, b, z)$ related to $M(a, b, z)$ via $M(a, b, z) = \Gamma(b)\mathbf{M}(a, b, z)$.

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
