# Peer review of "Eikonal amplitudes from curved backgrounds"

_SciPost Physics_

## Round 2 · Referee Report · Anonymous (Referee 1) · 2022-4-11

Report

The authors propose a general covariant framework to compute eikonal scattering amplitudes in terms of scattering in curved backgrounds. The method can be applied to general backgrounds, where the authors explore in detail the cases of Schwarzschild and Kerr backgrounds.

The quality of the work is impressive and the manuscript is very well-written and explained. The referee invites this work for publication in SciPost as it provides valuable input to the understanding of the eikonal scattering. However, there are minor review points the authors could consider before publication.

Requested changes

1 - Below Eq 1.1. The acronym ADM is not previously defined. Same for 'IR' below Eq 2.3 on page 4.

2 - Below Eq 2.3.The energy 'E' and momentum 'p' does not appear in this equation.

3- In Eq. 4.4, it seems that the effective source in momentum space dependence is missing $\hbar$ compared to the definition in Eq. 4.5.

4 - The interesting relation of Eq 4.21 is obtained after a small angle approximation. Does this equivalence still hold away from this approximation?

5 - Above Eq 5.7 it is said: "Using Eq 2.1 one obtains". Do the authors mean Eq 5.1?

6- Eq. 5.11. Should $\alpha$ be bold?

7 - Above Eq 5.14. Is the starting point for the factorization method Eq 5.10 instead of Eq 5.13?

8 - Eq 5.7. What is ${\cal C}_{\infty}$? Should it be ${\cal C}_L$?

9 - Caption of Figure 2. What are the contours ${\cal C}_{l}$ and ${\cal C}_{r}$?

10 - Figure 7. Can one comment on the wiggles at high-$q$. Is it just a numerical precision effect?

11 - Figure 8. Are the two curves plotted and do they precisely overlap?

  • validity: high
  • significance: high
  • originality: high
  • clarity: high
  • formatting: excellent
  • grammar: perfect

Author:  Andrea Cristofoli  on 2022-05-30  [id 2539]

(in reply to Report 1 on 2022-04-11)
Category:
answer to question

We thank the referee for the detailed reading and many helpful comments. We have responded to all of these below, indicating corresponding modifications to the draft.

1-We have defined these acronyms where they are introduced (ADM=Arnowitt-Deser-Misner, and IR=infrared).

2-We have removed the redundant/irrelevant definitions below equation (2.3).

3-We have added a $\hbar$ factor in (4.5)

4-The small angle approximation used in (4.14) is justified by the assumption of working at large distances from the source of the background, where the gravitational field is weak and the motion of an incoming particle only experiences small perturbations from a straight line. As for its validity, we expect this approximation to break down for strong fields and small distances from the source.

5-This equation reference has been corrected from (2.1) to (5.1).

6-The bold $\alpha$ is defined later, in equation (5.15) in terms of this $\alpha(s)$.

7-This equation reference has been corrected from (5.13) to (5.10).

8-Replaced $C_\infty$ by $C_L$ and added an underset symbol $L\to\infty$ for more precise notation in eq 5.17.

9-Those are $C_1$ and $C_2$, notation fixed.

10-The figure has been replaced with a figure with more points, so as to correct the precision effect. A comment has been added in the legend explaining the presence of these oscillations.

11-Two curves are plotted which precisely overlap. The legend has been clarified to emphasize this point.

Extra-The captions of fig 4 and 5 have been slightly expanded.

---

## Round 2 · Referee Report · Anonymous (Referee 2) · 2022-4-15

Strengths

1-Relevant and Timely
2-Scientifically sound
3-Well written

Weaknesses

1- Deals only with linear level results, while interest of the community

Report

This paper by Adamo, Cristofoli and Tourkine proposes an alternative derivation of the two-to-two eikonal amplitude in QFT in terms of a one-to-one amplitude in curved spacetime.

This paper is relevant and timely, considering the recent explosion in research linking the field of scattering amplitudes with the solution of the two-body problem in general relativity.

The research is scientifically sound, and the manuscript is well written, and so I recommend its publication in SciPost, perhaps after the authors have addressed the following minor questions / observations:

Requested changes

1-Linearization: In what setting would this go wrong if the linearization is not imposed? There is mounting evidence that the eikonal exponentiation will hold not only for linear level in the coupling constant, but also to higher orders. It is then my impression that the proposed formalism should also be able to encompass this. Is this correct? I believe the authors should put some emphasis on addressing this question, given that the program requires increasingly higher orders in the coupling constant.

2-Orthogonality: In eqs. (2.1) and (2.2) x_{\perp} is said to be orthogonal to {p_i}{\mu}. This is different from the definition in https://arxiv.org/pdf/2112.07556.pdf, which shares one author (AC) with the current manuscript, and where x is orthogonal to slightly different momenta. While there is no distinction at linear order in the coupling constant, such distinction plays an important role later, especially when considering spinning bodies. Perhaps the authors would consider making their notation consistent, or adding a clarification.

3-In section 5 a computation using harmonic coordinates has been performed, and it is stated that the result has to be coordinate independent. Would there be any advantage to use Kerr-Schild coordinates instead, where the graviton is exactly linear?

4-In section 5, it reads “The fact that the amplitude in the probe limit constrains the analogue arbitrary mass ratio result is a well known fact also noticed in observables such as the scattering angle”. This is only true up to 2PM. Beyond it, there are self-force corrections to the result.

5- From the abstract, it is stated that the amplitude presents a KLT factorization. This is confusing. While KLT’s origin is in string theory, it is more commonly understood (at least by myself, and a good fraction of the target audience of this paper) in its field theory sense of gravity=gauge^2, which is not the kind of factorization occurring here (unless it secretly is). Perhaps a clarification regarding in what sense this is like KLT would help the reader.

6- Slightly related to the previous point, the eikonal phase can also be obtained in other theories like QED, where the ladders also dominate, and even Yang-Mills, where this is no longer the case. Is there any way to obtain a similar derivation of the phase in such theories? Would it hold any relation to the KLT factorization in the integrals?

7- In the discussion’s “Spinning vs unspinning probe.” Reads “in the process the spins exponentiate and sum up in such a way that the end result depends only on the total spin a = a1+ a2”. This is only true at linear order in the coupling constant, which is apparently the scope of the paper, but not of the eikonal formalism. Also: “our proposal suggests that at least in a stationary approximation of the star { this amplitude will exhibit eikonal exponentiation”. I agree with this point. Some evidence in favor of this can be seen in the eikonal-like result in section 5 of https://arxiv.org/pdf/2102.10137.pdf. Other interesting questions are showing that having a neutron-star-like probe should also lead to (eikonal) exponentiation.

8-Also, they should consider the following style and typographic corrections Space-time -> Spacetime, time-like -> timelike, short q-> short-q Hevyside -> Heaviside Linearised->linearized (or the opposite, but keep it consistent)

9-Finally, I suggest (although this is completely optional) moving a fraction of the “Details of the calculation” in section 6.2 to an appendix, or at least streamlining it a bit, since in my opinion, it disrupts the flow of the paper.

  • validity: high
  • significance: good
  • originality: high
  • clarity: high
  • formatting: good
  • grammar: good

Author:  Andrea Cristofoli  on 2022-05-30  [id 2540]

(in reply to Report 2 on 2022-04-15)
Category:
answer to question

The referee correctly observes that our results only deal with the linearized gravitational field, which (as we show) is sufficient to recover the leading eikonal phase $\chi_1$. It would, of course, be interesting to push this program beyond the leading order; a natural conjecture is that keeping subleading powers of $G$ -- or equivalently, higher $r^{-1}$ corrections to the metric -- will produce sub-leading eikonal physics. However, at this stage we are not willing to speculate in greater detail on this point, leaving it to future research. As for the points raised by the referee,

1- In this paper, ``linearization" means that the gravitational field is considered only in the linear term in the coupling, while the results for the eikonal amplitudes contain all-orders loop contributions coming from the resummation of ladder and cross-ladder diagrams. In this sense, if linearization is not imposed, one can still prove that (4.21) holds. However, this requires a different calculation which we reserve to address in future works.

2- Regarding $x_{\perp}$ in (2.1) and (2.2): we say several times throughout the paper that we are only studying the eikonal exponentiation of the \emph{leading order} eikonal phase. (See, for example, the text before (2.1) and the use of the subscript ``1" in every expression containing the eikonal phase.) It should be clear then that the integration variable $x_{\perp}$ can only be the orthogonal projection of $x$ on the plane spanned by the incoming momenta $p_1$ and $p_2$. Moreover, we disagree on the comment that the spinning case would require more care in defining $x_{\perp}$ in (2.1). It is just an integration variable, and it remains the same in the spinless and spinning case. Spin dependence can only arise in the specific value of $x_{\perp}$ satisfying the stationary phase condition, if one were to apply a stationary phase approximation in the eikonal amplitude. It should be clear from the context if this is happening, so we prefer to leave the notation as it is.

3- Proposing the use of Kerr-Schild coordinates is natural, but it would require a more involved Fourier transform due to a non-trivial integration region. In this sense, the use of harmonic coordinates provides a concrete setting to evaluate the eikonal phase in the variable $x_{\perp}$ as the Fourier transforms to perform are simpler than those arising from Kerr-Schild coordinates.

4- Our statement was referring only to a probe particle and to the eikonal resummation of the leading order eikonal phase. We have underlined this point with a small change in the text.

5- Our use of the terminology ``KLT-like'' factorization is meant to invoke the holomorphic factorization inherent in the KLT formula. While in the context of string theory, this factorization is indeed related to double copy, here it is just a statement about the analytic structure of the eikonal amplitudes. We have tried to make this as explicit as possible in the body of the text and in the conclusion.

6-The answer is yes. The simpler case of scalar QED would provide the same KLT-like factorization properties seen in the gravitational context. (This emphasizes that the KLT-like factorization observed here is not a consequence of double copy.) We have commented on this in the conclusions. For Yang-Mills theory, the status is less clear as the eikonal approximation for non-abelian gauge theories is much richer: one cannot simply sum ladder and crossed ladder diagrams (since these come with distinct colour structures).

7- We have added some qualifiers to the discussion of this topic in the conclusion which hopefully clarify these subtleties.

8- We have implemented the referee's suggestions, uniformizing the the spelling of these terms through the draft.

9- We understand why the referee has made this suggestion, but our feeling is that the evaluation of the Kerr eikonal amplitude is one of the main results in this paper. As such, we prefer to keep the details of the computation intact in Section 6.

---

## Editorial Decision

resubmitted